# Hormone circuit explains why most HPA drugs fail for mood disorders and predicts the few that work

Tomer Milo (ID), Shiraz Nir Halber (ID), Moriya Raz, Dor Danan, Avi Mayo (ID) & Uri Alon (ID) ✉

## Abstract

Elevated cortisol in chronic stress and mood disorders causes morbidity including metabolic and cardiovascular diseases. There is therefore interest in developing drugs that lower cortisol by targeting its endocrine pathway, the hypothalamic–pituitary–adrenal (HPA) axis. However, several promising HPA-modulating drugs have failed to reduce long-term cortisol in mood disorders, despite effectiveness in other hypercortisolism conditions such as Cushing's syndrome. The reasons for these failures remain unclear. Here, we use a mathematical model of the HPA axis to demonstrate that the pituitary and adrenal glands compensate for drug effects by adjusting their functional mass, a feedback mechanism absent in Cushing tumors. Our systematic in silico analysis identifies two interventions targeting corticotropin-releasing hormone (CRH) as effective for lowering long-term cortisol. Other targets either fail due to gland mass compensation or harm other aspects of the HPA axis. We propose CRH-neutralizing antibodies and CRH-synthesis inhibitors as potential targets for reducing long-term cortisol in mood disorders and chronic stress. More generally, this study indicates that understanding the slow compensatory mechanisms in endocrine axes can be crucial to prioritize drug targets.

**Keywords** Dynamic Compensation; HPA Axis; Mood Disorders; Systems Endocrinology; Systems Pharmacology
**Subject Category** Computational Biology

## Introduction

Cortisol is a steroid hormone produced by the adrenal glands in response to physical or psychological stressors. It acts on almost every tissue in the body and mediates the stress response by regulating metabolism, cognitive functions and immune responses (Ramamoorthy and Cidlowski, 2016; Thau et al, 2024; Melmed et al, 2019).

Cortisol level is controlled by the hypothalamus–pituitary–adrenal (HPA) axis, a cascade of three hormones. In response to stressor inputs, the hypothalamus secretes corticotropin-releasing hormone (CRH). CRH stimulates the secretion of adrenocorticotropic hormone (ACTH) by corticotroph cells in the anterior pituitary, an effect enhanced by vasopressin (Aguilera et al, 2008; Antoni, 2017). ACTH in turn signals the adrenal cortex to secrete cortisol. Cortisol inhibits the production and secretion of the two upstream hormones, CRH and ACTH (Melmed et al, 2019), forming a negative feedback loop.

Prolonged elevated levels of cortisol, a condition known as hypercortisolism, can lead to a range of health issues (Melmed et al, 2019). These include weight gain, high blood pressure, diabetes, cardiovascular disease, osteoporosis, muscle weakness, thinning skin, increased bruising, slower wound healing, and mood changes. In addition, hypercortisolism can cause disruptions in sleep and memory, reduce libido, and compromise the immune system, making the body more susceptible to infections (Russell and Lightman, 2019; McEwen, 2017; Baum et al, 1999).

Hypercortisolism occurs in the context of chronic stress, such as that associated with low socioeconomic status (Dowd et al, 2009; Cohen et al, 2006) and also in mood disorders such as major depressive disorder (MDD) (Kennis et al, 2020) and bipolar disorder (BD) (Belvederi Murri et al, 2016; Milo et al, 2024). Hypercortisolism can also be caused by drugs and tumors. Drugs such as glucocorticoid steroids are cortisol analogs that cause the above-mentioned health issues upon prolonged treatment. Tumors in Cushing syndrome escape HPA regulation and cause elevated cortisol. Cushing's syndrome is often treated by tumor removal. When surgery is not possible in Cushing patients, the negative health effects of high cortisol are treated by cortisol-modulating drugs such as cortisol synthesis inhibitors and cortisol receptor antagonists (Nieman et al, 2015; Gilis-Januszewska et al, 2022).

Interestingly, Cushing's tumors and corticosteroid drugs frequently cause mood episodes (Pivonello et al, 2015; Sonino and Fava, 2001; Sonino et al, 1998; Fardet et al, 2012; Judd et al, 2014). This causal effect of elevated chronic cortisol on mood, as well as the strong association of stress and MDD, has raised the hope that cortisol-lowering drugs could improve mood disorder symptoms in MDD and BD (Thomson and Craighead, 2008; Wolkowitz and Reus, 1999; Menke, 2019).

It thus came as a disappointment when HPA medications effective in Cushing's syndrome failed in clinical trials for MDD and BD (see Table 1) (Ozbolt et al, 2013). These drugs include cortisol- and ACTH-synthesis inhibitors and glucocorticoid receptor (GR) antagonists. Drugs that lowered cortisol in Cushing syndrome failed to lower long-term cortisol in people with mood disorders, and some drugs such as GR antagonists even raised cortisol levels. Why HPA drugs that lower cortisol in Cushing syndrome show limited efficacy in mood disorders is not well

Department of Molecular Cell Biology, Weizmann Institute of Science, Rehovot 76100, Israel. ✉E-mail: uri.alon@weizmann.ac.il

**Table 1.  Long-term efficacy of HPA-related drugs in hypercortisolism conditions.**

| Class | Drug | Cushing syndrome (CS) | Psychiatric illness | Pre-clinical |
|---|---|---|---|---|
| Cortisol synthesis inhibitor | Metyrapone | Approved for CS by the EMA (Hinojosa-Amaya et al, 2019) | Mixed evidence for metyrapone efficacy in alleviating depressive mood in MDD (Jahn et al, 2004; McAllister-Williams et al, 2016) | |
| Cortisol synthesis inhibitor | Ketoconazole | Approved for CS by the EMA (Castinetti et al, 2021) | One study ($n = 20$) found antidepressant effects in hypercortisolemic but not in non-hypercortisolemic patients treated with ketoconazole for 4 weeks (Wolkowitz et al, 1999) Another study in MDD patients found no significant changes after 6 weeks of treatment ($n = 16$) (Malison et al, 1999) Limited evidence in short-term study ($n = 6$) for ketoconazole effect in BD patients (Brown et al, 2001) | |
| GR antagonist | Mifepristone | Mifepristone shown to be effective in treating CS (Fleseriu et al, 2012; Brown et al, 2020) Approved by FDA for treatment of CS since 2012. Serum cortisol levels remain unchanged or rise in response to GR antagonism (Katznelson et al, 2014) Suicidal ideation, depression and psychosis in CS patients were also resolved with mifepristone treatment (Chu et al, 2001; van der Lely et al, 1991; Nieman et al, 1985) | In a small, short-term study, the HDRS mean score of three out of four MDD patients decreased (Murphy et al, 1993) In psychotic depression (PD), mifepristone was tested for short-term only and showed mixed effects (Simpson et al, 2005; DeBattista and Belanoff, 2006; Block et al, 2018; Flores et al, 2006; Gallagher and Young, 2006) In bipolar disorder (BD), mifepristone showed no or short-term effect on mood symptoms (Young et al, 2004; Watson et al, 2012) Levels of cortisol rose during treatment (Murphy et al, 1993; Gallagher and Young, 2006; Young et al, 2004) | |
| CRH-receptor antagonist | R121919 | | A study on 24 MDD patients treated for 30 days with R121919 found significant reductions in depression and anxiety scores. Urinary-free and plasma cortisol decreased throughout the treatment (Zobel et al, 2000). | Treatment with DMP696 and R121919 decreased immobility time in the tail suspension test in mice (Nielsen et al, 2004) |
| CRH-receptor antagonist | Antalarmin | | | In monkeys, antalarmin has been successful in suppressing anxiety-associated behaviors (Habib et al, 2000) |
| CRH-receptor antagonist | CP-316,311 | | A clinical trial was terminated early due to no significant antidepressant effect (Binneman et al, 2008) | |
| CRH-receptor antagonist | Pexacerfont | | Pexacerfont did not demonstrate efficacy compared to placebo for the treatment of generalized anxiety disorder (Coric et al, 2010) | |
| Anti-CRH antibodies | CTRND05 | | | In mice, treatment with anti-CRH antibodies counteracts some of the effects of chronic variable stress (Futch et al, 2019) |
| Vasopressin 1B receptor ($V_{1B}$) antagonist | SSR149415 | | SSR149415 did not show significant benefits in GAD or two MDD trials compared to placebo (Griebel et al, 2012) | |
| Vasopressin 1B receptor ($V_{1B}$) antagonist | TS-121 | | TS-121 did not reach statistical significance after six weeks in a randomized, double-blind, placebo-controlled study for MDD (Kamiya et al, 2020) | |

GR glucocorticoid receptor, CS Cushing syndrome, HAM-D Hamilton Rating Scale for Depression, CGI Clinical Global Impressions, HDRS Hamilton Depression Rating Scale, PD psychotic depression, MDD major depressive disorder, BD bipolar disorder, FDA Food and Drug Administration, EMA European Medicines Agency, CRH corticotropin-releasing hormone, GAD generalized anxiety disorder.

understood. It is of interest to explore whether still untested HPA drugs might lower cortisol in the context of chronic stress and mood disorders.

Here we address this using a systems pharmacology approach by employing a recent advance in mathematical modeling of the HPA axis. The new model (Karin et al, 2020) updated the classical HPA model which works on the timescale of the hormone lifetimes, namely minutes to hours (Vinther et al, 2011). The classical model is thus not suited to address the timescale of weeks to months needed to assess chronic cortisol levels. The weeks–months timescale was introduced in the Karin et al mathematical model of the HPA axis by including changes over time in the functional mass of the endocrine glands (Karin et al, 2020). The larger the gland functional mass, the more hormone it secretes per unit input hormone. The gland size in the model is governed by well-characterized interactions which had not been previously considered on the systems level, namely that gland functional mass is regulated by the HPA hormones. CRH increases the functional mass of corticotrophs in the pituitary (Westlund et al, 1985; Asa et al, 1992) and ACTH serves as a growth factor for the cortisol-secreting cells in the adrenal (Swann, 1940; Lotfi and de Mendonca, 2016; Lopez et al, 2021). Since the cell turnover time in the pituitary and adrenal glands is on the order of months, glands grow and shrink on this slow timescale. The model shows how the gland masses adjust over months to buffer variation in physiological parameters, a property called *dynamical compensation* (Karin et al, 2016).

The gland mass model was tested and validated using longitudinal hair cortisol measurements in healthy individuals (Maimon et al, 2020) and in people with bipolar disorder (Milo et al, 2024) where it explained year-scale cortisol fluctuations. It was also validated and calibrated on a wide range of long-term phenomena such as hormone seasonality (Tendler et al, 2021), recovery from chronic stress (Karin et al, 2020) and addiction (Karin et al, 2021). The model was extended to understand the timescales of MDD (Ron Mizrachi et al, 2023) and BD (Milo et al, 2024). The concept of changes in gland mass was also adapted to the thyroid axis to explain dynamics of thyroid diseases (Korem Kohanim et al, 2022). These studies motivated us to use the HPA gland mass model to understand which HPA-modulating drugs might lower cortisol and which are destined to fail.

Here we use the gland mass mathematical model to systematically test in silico many possible HPA interventions for lowering long-term cortisol. We find that most drugs do not lower cortisol in an intact HPA axis due to the compensatory capacity of the gland masses, which change over weeks to completely nullify the drug effect. This compensation is broken in Cushing tumors which escape HPA regulation, explaining why Cushing's drugs are effective in lowering cortisol. We identify two CRH-associated drug targets that are expected to lower long-term cortisol in chronic stress conditions and mood disorders but not in Cushing's syndrome. These drugs also preserve all HPA hormone levels and response features. This study thus proposes that certain CRH-modulating drugs, such as neutralizing anti-CRH antibodies (Futch et al, 2019), may be effective to lower long-term cortisol in stress-related disorders. More generally, this study indicates that understanding the slow compensatory mechanisms in endocrine axes can be crucial in order to prioritize drug targets.

# Results

## Under chronic stress, CRH-associated interventions normalize long-term cortisol, whereas other interventions are compensated and fail

To search for strategies to lower long-term cortisol levels, we used the HPA gland mass model (Karin et al, 2020). The model incorporates the classical hormone cascade and negative feedback loop (Andersen et al, 2013). The gland mass interactions added to the classical HPA model are highlighted in bold in Fig. 1A. The model is agnostic to whether the functional mass grows by hypertrophy or hyperplasia.

In these interactions, the dynamics of the pituitary and adrenal functional masses are under the control of their upstream HPA hormone growth factors, CRH and ACTH, respectively. Since the model was found to be accurate in a wide range of clinical situations that undergo changes over months (Karin et al, 2020; Maimon et al, 2020; Tendler et al, 2021; Karin et al, 2021; Ron Mizrachi et al, 2023; Milo et al, 2024), we reasoned that it would be informative also for drug effects.

We tested in silico all possible points of intervention and asked whether they reduce cortisol in the long term. In this "circuit-to-target" approach, we systematically modeled each intervention's effect on the model parameters and looked for those that reduced steady-state cortisol (see "Methods"). In this way, we simulated agonists and antagonists for cortisol, ACTH or CRH receptors, neutralizing antibodies against each of the three hormones, and synthesis inhibitors of each of the three hormones (Fig. 1B).

We also tested possible combinations of interventions. We found that no new effective combinations arise apart from combinations of the single drugs described next.

We find that after a transient period of a few weeks, the HPA glands in the model change in size to compensate for most of the possible interventions. For example, blocking the glucocorticoid receptor (GR) with GR antagonists led to an increase in the adrenal cortex mass. The increased adrenal mass generated higher levels of cortisol that *precisely negated* the reduction in GR binding to cortisol (Fig. 1B, bottom right panel). This precise compensation happens no matter what the dose of the drug. Thus, a GR antagonist has no net effect on GR signaling after a transient period of a few weeks. The rise in cortisol agrees with observations from clinical trials using the GR antagonists (Table 1) (Murphy et al, 1993; Gallagher and Young, 2006; Young et al, 2004). This effect would not be seen in the classical HPA model.

Similarly, ACTH-receptor antagonists did not affect cortisol levels after a few weeks because the adrenal gland mass grew to compensate precisely for the inhibition. Synthesis inhibitors of cortisol and ACTH likewise had only a transient effect of a few weeks, which vanished once gland masses changed to fully compensate for the intervention.

It can be shown mathematically why the compensatory properties of the circuit prevent most parameter changes from altering cortisol steady state in the long term (see Eqs. (7)–(16) in "Methods"). This is because cortisol steady state, $cortisol_{st}$, is robust to changes in most of the circuit parameters. It depends on

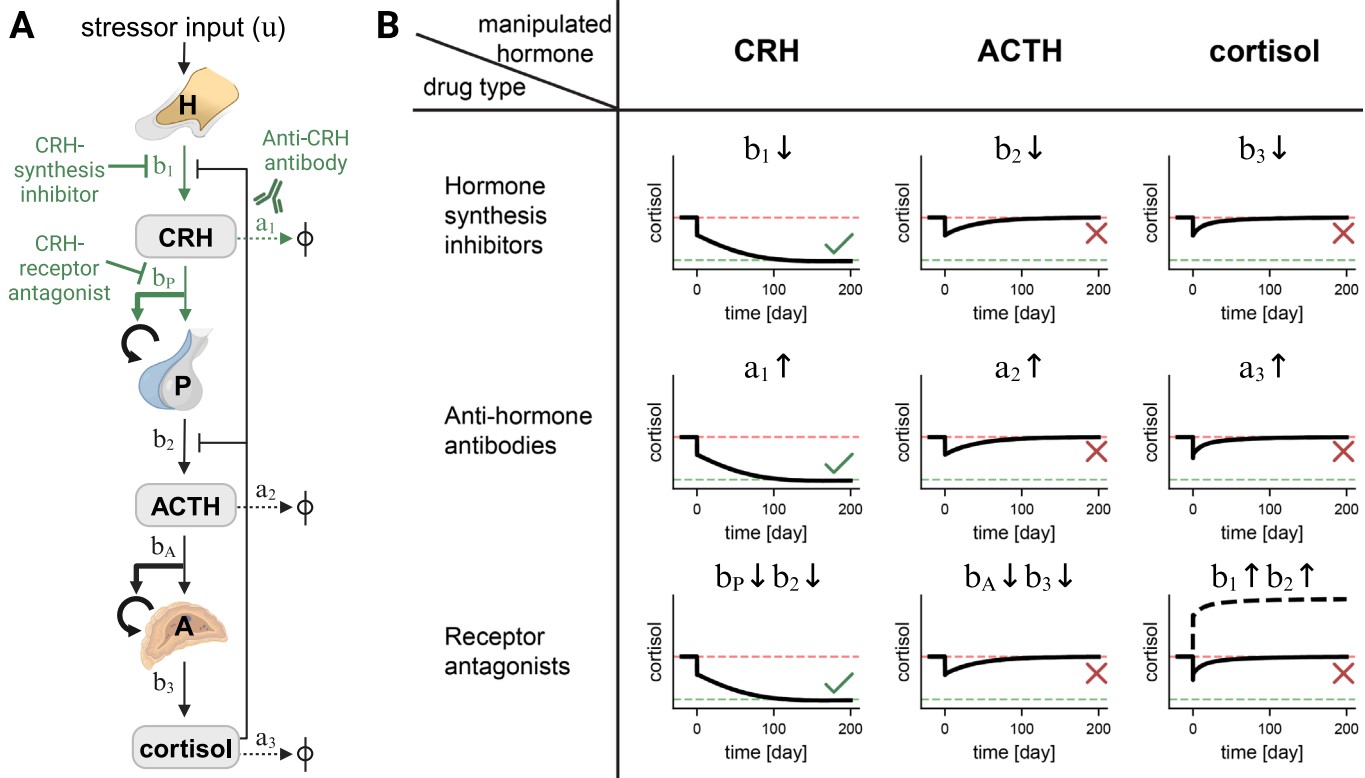

**Figure 1. A circuit-to-target approach to lowering long-term cortisol levels in chronic stress conditions points to a few drug targets that can be effective, whereas most are predicted to fail.**

(A) The HPA circuit diagram. The hypothalamus H secretes CRH at rate $b_1$ in response to a stressor input $u$. CRH causes the pituitary P to secrete ACTH at rate $b_2$ and to grow in functional mass at rate $b_P$. ACTH signals the adrenal gland to secrete cortisol at rate $b_3$ and to grow in functional mass at rate $b_A$. The hormone removal rates are $a_1$, $a_2$ and $a_3$ for CRH, ACTH and cortisol, respectively. Thick arrows indicate the interactions added in the Karin et al model that affect gland sizes on the scale of months. The drugs predicted to be effective and their points of intervention are illustrated in green. (B) Simulations of HPA interventions. Simulations began with an elevated level of cortisol (marked by horizontal dashed red lines) due to chronic stress input and were run for 200 days. A single simulated drug was administered at time zero. The parameter influenced by each drug and its direction of change is indicated in each panel (see "Methods"). As an example, note that cortisol receptor antagonists inhibit the negative feedback of cortisol on CRH and ACTH synthesis and thus effectively increase CRH and ACTH production rates, $b_1$ and $b_2$, respectively. Interventions that succeeded in reducing cortisol to a normal level (horizontal dashed green lines) are marked with a green check mark, whereas those that failed are marked with a red X. In the case of cortisol receptor antagonists, the dashed black line indicates cortisol level, and the continuous black line is the cortisol net effect through GR signaling on target cells after accounting for the receptor-blocking effect by the antagonists.

only a few parameters that are associated with CRH signaling:

$$cortisol_{st} = \frac{b_1 b_P}{a_1 a_P} u \qquad (1)$$

This equation also reveals which interventions can lower cortisol steady state. According to Eq. (1), cortisol steady state can be reduced by CRH-related interventions (Fig. 1B, left column). First, one can lower CRH production rate $b_1$ by using CRH-synthesis inhibitors. Second, one can lower the effect of CRH on pituitary corticotroph cell growth rate $b_P$. This can be done by inhibiting the CRH receptor on the pituitary corticotrophs using a receptor antagonist (Zobel et al, 2000). Another way to lower cortisol is to increase the CRH removal rate $a_1$, for example by using antibodies that bind and neutralize CRH. An anti-CRH antibody has been shown to suppress the HPA axis in stressed mice (Futch et al, 2019). Such drugs may be candidates for treating chronic stress conditions.

According to Eq. (1), as a general guideline, treating cortisol levels that are x-fold higher than baseline requires a drug dose that alters the relevant parameter (e.g., CRH production or removal rate) by a similar x-fold.

The model further predicts that lowering the input $u$, the stress signal in the brain communicated to the hypothalamus and leading to CRH secretion, can also lower cortisol steady state. This might relate to psychotherapy, exercise, and other lifestyle interventions that reduce stress (Benson and Klipper, 1975). Finally, increasing the pituitary corticotroph removal rate $a_P$ should also lower cortisol steady state.

All other drugs that target ACTH or cortisol, are predicted to have only a transient effect lasting a few weeks. This includes receptor antagonists, hormone production inhibitors, or anti-hormone antibodies. Such drugs do not affect cortisol steady state and thus fail to lower cortisol levels in the long term (Fig. 1B, middle and right columns). The classical HPA model with nonadjustable glands predicts that cortisol steady state would

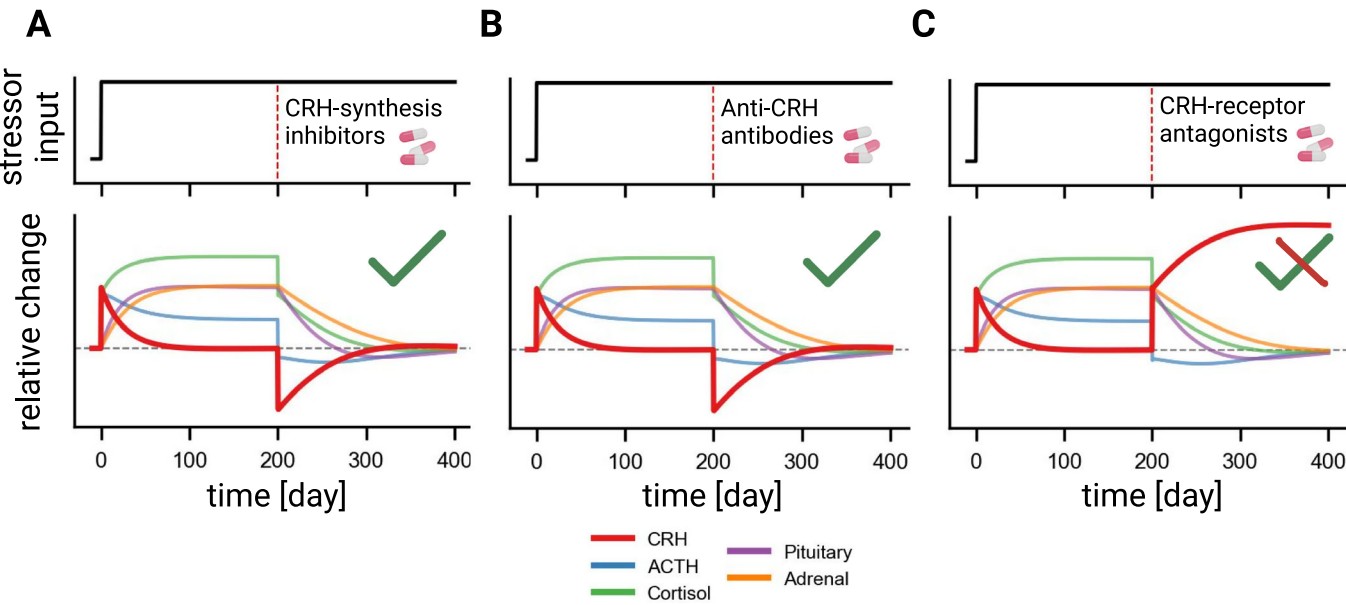

**Figure 2. Two of the CRH-targeting drugs normalize all hormone and gland masses in chronic stress, whereas one raises CRH.**

Chronic stress was simulated by increasing the input $u$ at time zero and keeping it elevated for the entire simulation (upper panels). The drug was administered 200 days later (red vertical dashed line). The simulated HPA responses to CRH-synthesis inhibitors (**A**), anti-CRH antibodies (**B**), and CRH-receptor antagonists (**C**) are presented in the lower panels. CRH dynamics are highlighted in red. The gray horizontal dashed line indicates the healthy baseline.

depend on ACTH and cortisol parameters (see Fig. EV1), and thus cannot predict these compensation effects.

Importantly, these conclusions on drug effect do not depend on the HPA model parameter values, since they can be analytically derived from the model's steady-state solution. They are thus a robust prediction.

## CRH-synthesis inhibitors and anti-CRH antibodies normalize all HPA hormones and glands

Next, we asked whether the effective drugs normalize not only cortisol but also the levels of the other HPA components. We find that two of the CRH interventions, namely CRH-synthesis inhibitors and anti-CRH antibodies normalize the entire HPA axis—all hormone levels and gland masses return to normal (Fig. 2A,B).

In contrast, CRH-receptor antagonists are predicted to reduce long-term cortisol but increase long-term CRH levels (Fig. 2C). Higher CRH steady state is due to the pituitary's integral feedback loop that locks CRH levels (Alon, 2023; Karin et al, 2020). Physiologically, CRH-receptor antagonists reduce the gland mass effect of CRH on pituitary cells, and thus more CRH is needed to keep the pituitary at a fixed size.

## CRH interventions preserve acute responses to relative stressors

Given the predicted efficacy of CRH-synthesis inhibitors and anti-CRH antibodies in normalizing the HPA axis under chronic stress conditions, we next asked whether these interventions also preserve acute stress responses. It is important to evaluate the acute stress response, in order to avoid drugs that normalize the axis in the long

term, but impair short-term responses on the timescale of hours which are critical to successful stress responses.

We simulated the HPA model with acute stressors in the form of short-term pulses of input $u$ above the baseline input (Fig. 3, top row). The first acute stressor input was simulated at the healthy baseline and serves as a reference for comparison. The second acute stressor pulse was simulated during the chronic stress period with drug treatment (Fig. 3, top row).

We find that the HPA acute stress response under the CRH-targeting medications were identical to the normal healthy responses (Fig. 3). In line with this prediction, a proper acute stress response was observed in mice treated with anti-CRH antibodies after two weeks of chronic variable stress (Futch et al, 2019). We thus conclude that CRH-synthesis inhibitors and anti-CRH antibodies are both predicted to normalize HPA components in the long term and preserve acute stress responses.

## CRH interventions have a dose-dependent response in the model

We computed the effects of drug doses by varying the relevant model parameter, where zero dose means no change in the parameter and high doses mean large changes in the parameter. We find that both candidate interventions for lowering cortisol—CRH-synthesis inhibitors and CRH-blocking antibodies—cause a dose-dependent reduction of steady-state cortisol (Fig. 4A). This indicates that putative treatment may require finding the appropriate dose to return the patients to their normal cortisol baseline range. Other drug candidates have no effect on long-term cortisol steady state (Fig. EV2).

At all doses, the steady states of CRH and ACTH remain normal (Fig. 4B,C). The acute stress response, defined

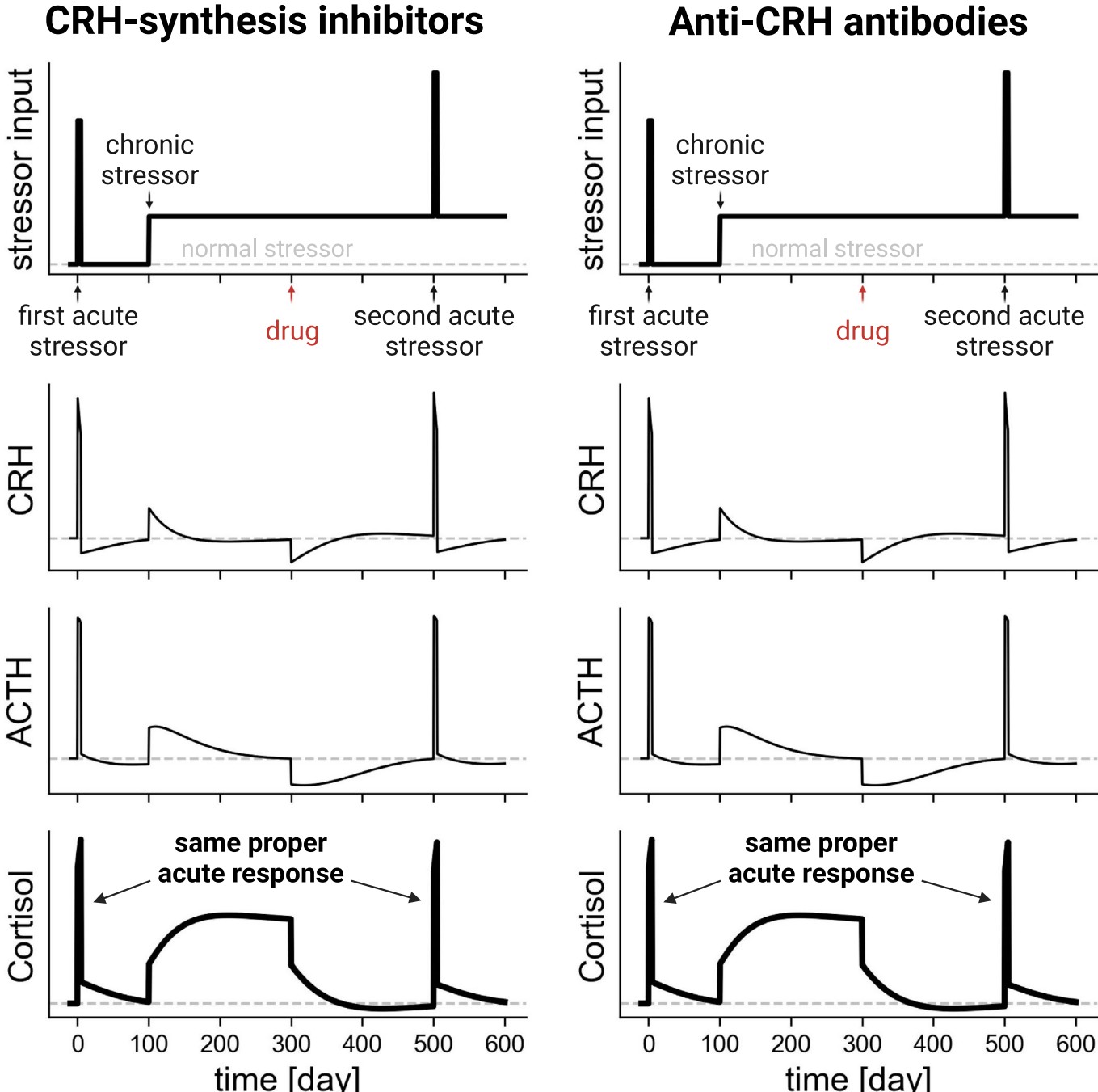

**Figure 3.  HPA responses to acute stress are preserved during CRH-related interventions.**

Acute stress was simulated as a short-term pulse above a baseline (top row). The first acute stress was introduced during the healthy period, before the chronic stress. The second acute stress was simulated during the chronic stress period, several months after drug treatment began. The response of HPA hormones to these stressors for CRH-synthesis inhibitors (left column) or for anti-CRH antibodies (right column) is depicted in the bottom three rows. The dashed lines indicate the healthy normal state.

as peak cortisol upon acute stress input relative to steady-state cortisol, is dose-dependent (Figs. 4D and EV3). At a dose that returns cortisol to the normal, the acute response is also normalized.

We also tested the effects of abrupt treatment cessation. For both CRH interventions, stopping treatment led to a rapid return to hypercortisolemia (Figs. 4E,F and EV4).

**Endogenous Cushing's syndrome responds to different drug targets due to loss of gland compensation**

We next aimed to understand why HPA-targeting drugs that failed in mood disorder trials succeed in treating hypercortisolism in Cushing's syndrome. We investigated the two main classes of Cushing's syndrome—tumors that produce an excessive amount of

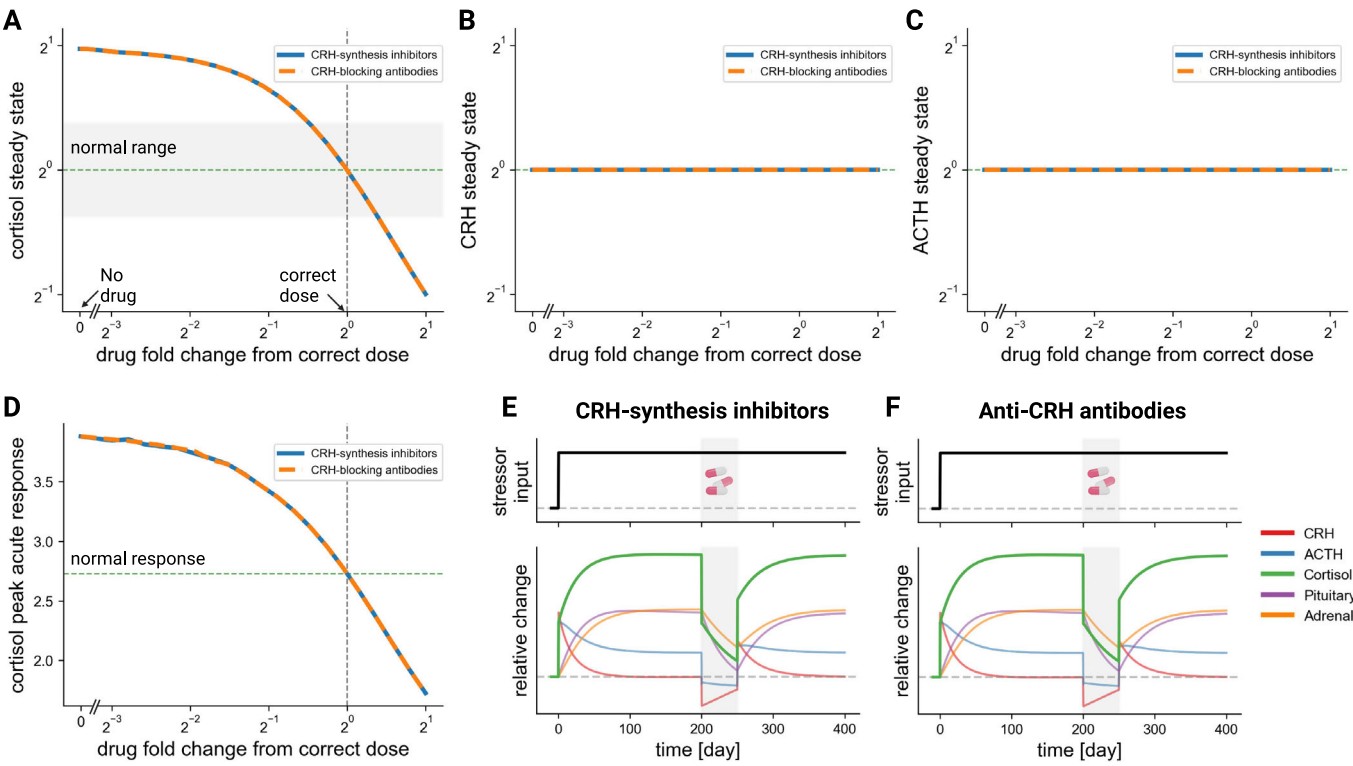

**Figure 4. Predicted effective interventions have a dose-dependent effect on cortisol, and cortisol abruptly rises when treatment is ceased.**

(A) Cortisol steady state in the model as a function of dose of CRH-synthesis inhibitors and CRH-blocking antibodies. (B, C) The same drug doses have no effect on ACTH (B) and CRH (C) steady-state levels. (D) Cortisol peak response to acute stress relative to steady state as a function of drug doses. (E, F) HPA dynamics upon cessation of CRH-synthesis inhibitors (E) and anti-CRH antibodies (F) after 50 days of treatment.

ACTH (Cushing's disease and ectopic ACTH syndrome) and tumors that overproduce cortisol (adrenal gland tumors). The ACTH-dependent cases are more common (Lacroix et al, 2015). Importantly, the tumors escape the HPA feedback loops—high cortisol does not suppress hormone production in the mutant tumor cells.

Cushing's tumors are often treated with surgery. However, tumor location, patient health status, or patient preference can favor medication rather than surgery. Drugs are also sometimes used temporarily to control the deleterious effects of cortisol before surgery or while waiting for the effects of radiation therapy (Nieman et al, 2015; Gilis-Januszewska et al, 2022).

To analyze HPA drugs in Cushing's syndrome, we follow the analysis of (Raz et al, 2023). We modeled the tumor secretion rates by adding the appropriate production terms to the equations ("Methods", Eqs. (18) and (26)). We modeled tumor growth by a logistic function (Vaghi et al, 2020; Murphy et al, 2016) (Fig. 5A,B, top panels). Our findings are not sensitive to the exact functional form of tumor growth. In both adrenal and pituitary adenomas, as long as the tumor is below a certain threshold of secretion rate (vertical dashed gray lines, Fig. 5A,B), the HPA glands compensate to keep the hormones at normal levels (see "Methods") (Raz et al, 2023). During this stage, the tumor is predicted to be subclinical with no overt symptoms.

In the case of an ACTH-overproducing tumor, the healthy pituitary corticotroph mass shrinks (Fig. 5A, middle panel). Once

the tumor secretion crosses a threshold, the pituitary is too small to compensate, and ACTH rises. An increase in ACTH leads to an increase in cortisol and adrenal mass increase. In this regime, cortisol steady state becomes dependent on ACTH and cortisol parameters (see Methods). Thus, ACTH- or cortisol-targeting drugs can reduce cortisol levels as clinically observed (Fig. 5C, left table). These include ACTH or cortisol synthesis inhibitors, ACTH or cortisol receptor antagonists, and anti-ACTH- or anti-cortisol-blocking antibodies.

The model also predicts that if the pituitary corticotrophs were not completely suppressed at the time of drug administration, the entire HPA axis could potentially recover. This recovery happens because the drugs effectively increase the subclinical–clinical threshold and move the patient back into the subclinical regime (see "Methods"). Although all ACTH- and cortisol-targeting drugs lower long-term cortisol (Fig. 5C, left table), only ACTH-synthesis inhibitor and anti-ACTH-blocking antibodies are predicted to normalize all HPA components (Fig. EV5).

In the case of an adrenal tumor, the healthy (non-tumor) adrenal glands shrink to compensate for the tumor's overproduction of cortisol (Kong et al, 2020; Park et al, 2016) (Fig. 5B, middle panel). Once the healthy adrenal tissue is too small to compensate, cortisol level rises and inhibits the upstream hormones. As a result, the pituitary corticotroph functional mass shrinks as well. At this stage, cortisol level is determined by the adrenal tumor alone (see "Methods"). Therefore, only drugs that manipulate cortisol

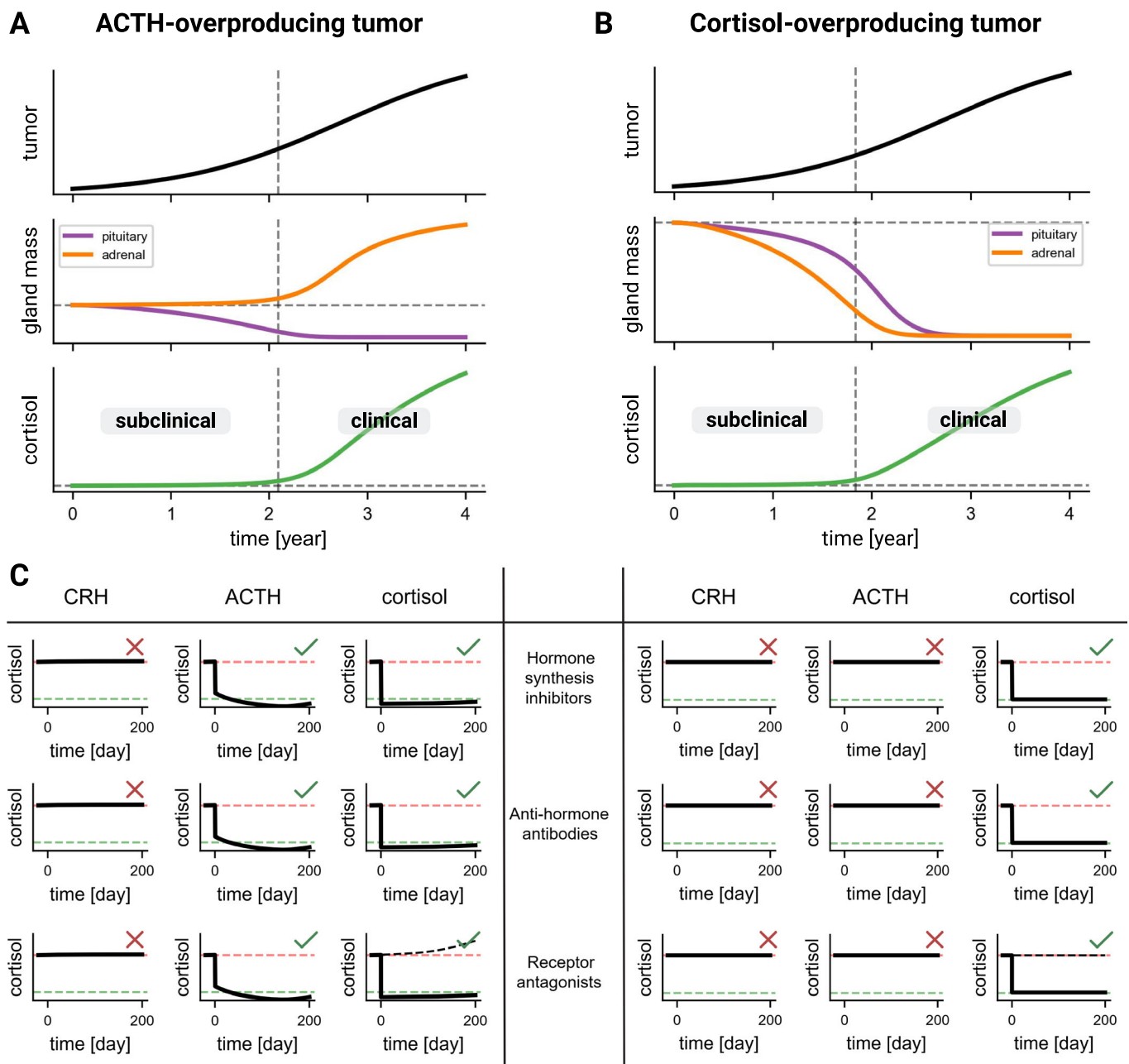

**Figure 5. Dynamics of natural and medicated Cushing's syndrome show that different sets of drugs are effective.**

(A, B) Simulations of ACTH-overproducing pituitary adenoma (A) and cortisol-overproducing adrenal adenoma (B). The HPA glands change their mass (middle panel) as the tumor grows (upper panel) and thus maintain homeostatic levels of cortisol (bottom panel) up to a threshold tumor secretion rate (vertical gray dashed line). Below this threshold cortisol levels do not rise—the subclinical phase. Above this threshold the glands cannot compensate and cortisol levels rise—the clinical phase. Gland mass in the middle panel corresponds to non-tumor functional mass, not to be confused with the tumor mass. (C) Simulations of potential HPA-modulating drugs to treat pituitary adenoma (left table) and adrenal adenoma (right table). Red X marks a drug predicted to fail in treating Cushing hypercortisolism, and a green check mark marks drugs predicted to be effective. Red and green horizontal dashed lines indicate abnormal and normal cortisol levels. In simulations of cortisol receptor antagonists, cortisol levels are in a black dashed line and the net effect of cortisol on target cells, after the antagonist effect, is in black continuous line.

parameters—cortisol synthesis inhibitors, anti-cortisol antibodies or cortisol receptor antagonists—should work to lower long-term cortisol levels (Fig. 5C, right table). Similar to the ACTH-overproducing tumor, these cortisol-targeting drugs are predicted to increase the subclinical–clinical threshold and recover all HPA components (Fig. EV5).

Interestingly, Mifepristone, a GR antagonist, resolved psychosis and depression symptoms in Cushing's patients (van der Lely et al, 1991; Chu et al, 2001) but was not effective in MDD patients without Cushing's syndrome (Table 1), aligning with our results.

Note that in Cushing scenarios, drug effects are predicted to work on the timescales of hours since the immediate effects do not

depend on the slow gland timescales. However, hypoadrenalism is predicted to occur for months due to the time it takes gland functional mass to return to normal (Fig. EV5). This aligns with observations of adrenal insufficiency with a recovery time of many months after successful surgery of pituitary and adrenal tumors (Cui et al, 2023; Graber et al, 1965). Notably, different sets of drugs work for the two types of Cushing syndrome, and none of these are predicted to lower cortisol in the HPA axis without a tumor.

## Discussion

We present a systems pharmacology approach to discover HPA drug targets to lower long-term cortisol in mood disorders and chronic stress conditions. We used a validated mathematical model that takes into account compensatory changes in endocrine gland functional mass (Karin et al, 2020). Most drugs are predicted to fail due to compensation by gland changes over weeks. We identify two drug targets that are predicted to lower cortisol—CRH-synthesis inhibitors and anti-CRH antibodies. These interventions also normalize all other HPA hormones and maintain a proper acute stress response. Other interventions that target ACTH or cortisol directly are predicted to transiently lower cortisol for several weeks, followed by a return of cortisol to its aberrant baseline. Different drug targets are effective in Cushing's tumors which bypass the normal HPA feedback loops. Thus, we conclude that CRH-synthesis inhibitors or neutralizing antibodies cannot be compensated by the HPA axis, and are candidates to lower cortisol in mood disorders and in chronic stress.

These findings explain the failure of clinical trials that attempted to use HPA-modulating drugs such as GR antagonists and cortisol synthesis inhibitors to treat mood disorders (Table 1). These same drugs, such as the GR antagonist mifepristone, work well in improving symptoms in Cushing's syndrome patients including mood symptoms (Nieman et al, 2015). We propose that the failure of Cushing drugs in mood disorders can be understood by considering the ability of the HPA glands to compensate for these interventions by changing their functional mass. For example, administering GR antagonists reduces the cortisol negative feedback on the upstream hormones, CRH and ACTH. Therefore, their levels rise and lead to higher secretion of cortisol and growth of the glands to exactly negate the GR antagonist effect on the receptor. In Cushing's syndrome, due to tumors that have lost the HPA feedback loops, ACTH- and cortisol-targeting drugs work because the glands lose their ability to compensate for the drug effects. Note that the classical model, with no changes in gland masses, cannot explain the compensation of drugs by the HPA axis.

To systematically test all possible points of intervention in the HPA axis we extended the Karin et al mathematical model for the HPA axis which incorporates the slow timescale dynamics of the endocrine glands (Karin et al, 2020; Raz et al, 2023). We used numerical simulations to analyze HPA dynamics under different medication regimes. We also analytically solved the steady states of the system. This approach rigorously defines the potential points of intervention. The qualitative conclusions on which drugs work does not depend on the model parameter values.

Our model is a simplified representation of the complex HPA biology. One important factor not explicitly considered in the model is the contribution of vasopressin to the axis. Vasopressin

potentiates the CRH-dependent release of ACTH from pituitary corticotrophs by acting on the V1b receptor (V1bR) (Aguilera et al, 2008; Antoni, 2017). Including this hormone explicitly is beyond the current scope. However, a simple analysis indicates that the effect of elevated vasopressin can be modeled by increasing the ACTH secretion parameter $b_2$. This suggests that vasopressin V1b receptor antagonists should have effects similar to inhibitors of ACTH production. As such, vasopressin receptor antagonists should be compensated by the HPA axis without long-term effects on cortisol. Accordingly, V1bR antagonists did not show statistically significant efficacy in clinical trials for major depressive disorder and generalized anxiety disorder (Griebel et al, 2012; Chaki, 2021; Kamiya et al, 2020). However, vasopressin may have additional relevant effects on the HPA axis and the central nervous system which warrant a more detailed modeling analysis.

An additional dynamical aspect that we neglected but could be important to investigate is the diurnal pulsatile secretion pattern of CRH and ACTH (Walker et al, 2010; Young et al, 2001). Future work might also consider modeling precise drug pharmacokinetics and a more accurate model of tumor growth. A more complete but more complex model can include crosstalk with other endocrine systems such as the thyroid axis (Seshadri et al, 2022; Singh and Sundaresh, 2022). Future work can also consider treatment for other HPA-related conditions such as Addison's disease, congenital adrenal hyperplasia (CAH) and post-traumatic stress disorder (PTSD).

In conclusion, our circuit-to-target approach explains why attempts to lower long-term cortisol using most HPA-targeting drugs are doomed to fail due to compensation by functional gland mass changes. We find that only a few drug targets can lower cortisol in the long term while preserving all other hormone levels. Predicted effective drugs include inhibitors of CRH synthesis or drugs that increase CRH removal such as anti-CRH antibodies. Such drugs may hold promise to treat cortisol-related mood disorders (Ron Mizrachi et al, 2023; Belvederi Murri et al, 2016; Kennis et al, 2020; Milo et al, 2024) and to mitigate the deleterious health effects of cortisol in those suffering from chronic stress. More generally, this study indicates that understanding the slow compensatory mechanisms in endocrine axes can be crucial in order to prioritize drug targets.

## Methods

### Reagents and tools table

| Reagent/resource | Reference or source | Identifier or catalog number |
|---|---|---|
| **Software** | | |
| Code for simulating HPA-targeting drugs | https://github.com/tomermilo/hpa-drugs | |

### Methods and protocols

#### The HPA gland mass model

To test the effect of HPA-targeting drugs in chronic stress conditions and Cushing tumors, we used the HPA gland mass

**Table 2. Definitions of the HPA mathematical model variables and parameters.**

| Dynamic variable | Definition |
|---|---|
| $x_1$ | Corticotropin-releasing hormone (CRH) |
| $x_2$ | Adrenocorticotropic hormone (ACTH) |
| $x_3$ | Cortisol hormone |
| $P$ | Pituitary corticotroph functional mass |
| $A$ | Adrenal functional mass |
| **Parameter** | **Definition** |
| $b_1, b_2, b_3$ | CRH, ACTH, cortisol production rates |
| $a_1, a_2, a_3$ | CRH, ACTH, cortisol degradation rates |
| $b_P, b_A$ | Growth rates of pituitary corticotroph and adrenal functional mass |
| $a_P, b_P$ | Removal rates of pituitary corticotroph and adrenal functional mass |
| $K_{GR}$ | Glucocorticoid receptor dissociation constant |
| **Intervention** | **Definition** |
| $u$ | Stressor input |
| $I_1, I_2, I_3$ | CRH, ACTH, and cortisol synthesis inhibitors |
| $C_1, C_2, C_3$ | CRH, ACTH, and cortisol receptor antagonists |
| $A_1, A_2, A_3$ | CRH, ACTH, and cortisol neutralizing antibodies |

model (Karin et al, 2020) and added explicit parameters for each possible intervention (see Table 2 for a description of model variables and parameters):

$$\frac{dx_1}{dt} = I_1 b_1 \cdot GR(C_3 x_3) \cdot MR(C_3 x_3) u - A_1 a_1 x_1 \tag{2}$$

$$\frac{dx_2}{dt} = I_2 b_2 C_1 x_1 GR(C_3 x_3) P - A_2 a_2 x_2 \tag{3}$$

$$\frac{dx_3}{dt} = I_3 b_3 C_2 x_2 A - A_3 a_3 x_3 \tag{4}$$

$$\frac{dP}{dt} = P(b_P C_1 x_1 - a_P) \tag{5}$$

$$\frac{dA}{dt} = A(b_A C_2 x_2 - a_A) \tag{6}$$

In response to an input stressor, $u$, the hypothalamus secretes CRH, $x_1$, at a rate $b_1$. CRH stimulates the corticotrophs at the pituitary, $P$, to secrete ACTH, $x_2$, at a rate $b_2$. ACTH signals the adrenal cortex of the two adrenal glands, whose total functional mass is $A$, to secrete cortisol, $x_3$, at a rate $b_3$. Cortisol inhibits the production of CRH and ACTH through the mineralocorticoid and glucocorticoid receptors, given by $MR(x) = \frac{1}{x}$; $GR(x) = \frac{1}{\left(\frac{x}{K_{GR}}\right)^n + 1}$, respectively. CRH, ACTH and cortisol are removed at rates $a_1$, $a_2$ and $a_3$, respectively. The gland mass model includes the effects of CRH on the pituitary functional mass ($b_P x_1$) and of ACTH on the adrenal functional mass ($b_A x_2$).

We added an explicit parameter for each possible intervention: hormone-synthesis inhibitors, $I_i < 1$, reduce hormone production rate, $b_i$, and does not affect anything else explicitly because $I_i$ appears only multiplying $b_i$; anti-hormone-blocking antibodies, $A_i > 1$, increase hormone removal rate, $a_i$; hormone-receptor antagonists or agonists, $C_i$, modulate the effect of hormone $x_i$ on its corresponding receptor and thus are coupled to obtain the hormone net effect, $C_i x_i$. CRH-receptor antagonists or agonists, $C_1$, affect both ACTH production rate, $b_2$, and corticotroph growth rate, $b_P$. Similarly, ACTH-receptor antagonists or agonists, $C_2$, effectively modulate cortisol production rate, $b_3$, and adrenal growth rate, $b_A$. Agonists or antagonists to cortisol receptors affect CRH and ACTH production rates, $b_1$ and $b_2$, because cortisol inhibits CRH and ACTH synthesis in the hypothalamus and the pituitary, correspondingly.

### Analytical solutions of the HPA model steady state

**Acute and chronic stress conditions**: Physiological and psychological stressor inputs cause the secretion of CRH. We denote stressor input magnitude to the hypothalamus by $u$. Acute and chronic stressors are modeled as a short or a prolonged fold change increase from the baseline input, $u = 1$. Input stressors that are very large induce high levels of cortisol that saturate the GRs, therefore we assume $x_3 \gg K_{GR}$. Under this approximation and the GR Hill coefficient, $n = 3$, $GR(x) = \frac{1}{\left(\frac{x}{K_{GR}}\right)^3 + 1} \simeq \left(\frac{K_{GR}}{x}\right)^3$. We also validated our conclusions with numerical simulations, relieving this assumption. In this regime the model and its steady state, denoted with $st$ subscript, are:

$$\frac{dx_1}{dt} = I_1 b_1 \left(\frac{K_{GR}}{C_3 x_3}\right)^3 \frac{1}{C_3 x_3} u - A_1 a_1 x_1 \qquad x_{1,st} = \frac{a_P}{C_1 b_P} \tag{7}$$

$$\frac{dx_2}{dt} = I_2 b_2 C_1 x_1 \left(\frac{K_{GR}}{C_3 x_3}\right)^3 P - A_2 a_2 x_2 \qquad x_{2,st} = \frac{a_A}{C_2 b_A} \tag{8}$$

$$\frac{dx_3}{dt} = I_3 b_3 C_2 x_2 A - A_3 a_3 x_3 \qquad x_{3,st} = \frac{1}{C_3} \sqrt[4]{\frac{C_1 I_1 K_{GR}^3 b_1 b_P u}{A_1 a_1 a_P}} \tag{9}$$

$$\frac{dP}{dt} = P(b_P C_1 x_1 - a_P) \qquad P_{st} = \frac{A_2 a_2 b_P a_A}{C_2 I_2 b_2 a_P b_A} \left(\frac{C_1 I_1 b_1 b_P u}{A_1 a_1 a_P K_{GR}}\right)^{3/4} \tag{10}$$

$$\frac{dA}{dt} = A(b_A C_2 x_2 - a_A) \qquad A_{st} = \frac{A_3 a_3 b_A}{C_3 I_3 b_3 a_A} \sqrt[4]{\frac{C_1 I_1 K_{GR}^3 b_1 b_P u}{A_1 a_1 a_P}} \tag{11}$$

Under the approximation of $x_3 \ll K_{GR}$, the GR is not appreciably activated ($GR(x) \simeq 1$) and the steady-state solution is:

$$\frac{dx_1}{dt} = I_1 b_1 \frac{1}{C_3 x_3} u - A_1 a_1 x_1 \qquad x_{1,st} = \frac{a_P}{C_1 b_P} \tag{12}$$

$$\frac{dx_2}{dt} = I_2 b_2 C_1 x_1 P - A_2 a_2 x_2 \qquad x_{2,st} = \frac{a_A}{C_2 b_A} \tag{13}$$

$$\frac{dx_3}{dt} = I_3 b_3 C_2 x_2 A - A_3 a_3 x_3 \qquad x_{3,st} = \frac{C_1 I_1 b_1 b_P}{C_3 A_1 a_1 a_P} u \tag{14}$$

$$\frac{dP}{dt} = P(b_P C_1 x_1 - a_P) \qquad P_{st} = \frac{A_2 a_2 b_P a_A}{C_2 I_2 b_2 a_P b_A} \tag{15}$$

$$\frac{dA}{dt} = A(b_A C_2 x_2 - a_A) \qquad A_{st} = \frac{C_1 I_1 A_3 b_1 a_3 b_P b_A}{A_1 C_3 I_3 a_1 b_3 a_P a_A} u \tag{16}$$

In both cortisol regimes, steady state cortisol $x_{3,st}$ can be reduced only by CRH-synthesis inhibitors, $I_1 < 1$; CRH-neutralizing antibodies, $A_1 > 1$; and CRH-receptor antagonists, $C_1 < 1$. GR antagonists, $C_3 < 1$, are predicted to increase cortisol levels in this scenario. However, the net effect of cortisol on target cells, $C_3 x_3$, after accounting for the competing antagonists ($C_3$), cancels out. The steady-state solution also shows that CRH-receptor antagonists ($C_1 < 1$) are predicted to increase CRH levels, and are thus less favorable.

These results do not hold in a nonadjustable gland mass model (see "Methods" and Fig. EV1).

**Pituitary adenoma (Cushing's disease):** To model a pituitary adenoma, we use the approach developed by (Raz et al, 2023). We add the tumor ACTH secretion capacity, $T_P(t)$, to Eq. (3). We assume the tumor increases with time according to a logistic rule. The exact dynamics of tumor growth does not influence our conclusions because, as demonstrated below, the bifurcation of the system's steady state depends on the tumor crossing a specific threshold and not on its exact temporal dynamics.

To study the disease progression we start with normal levels of cortisol which on average obey $x_3 << K_{GR}$. In this limit $GR(x) \simeq 1$ and the system becomes:

$$\frac{dx_1}{dt} = I_1 b_1 \frac{1}{C_3 x_3} u - A_1 a_1 x_1 \quad x_{1,st} = \frac{a_P}{C_1 b_P} \tag{17}$$

$$\frac{dx_2}{dt} = I_2 b_2 C_1 x_1 P + I_2 T_P(t) - A_2 a_2 x_2 \quad x_{2,st} = \frac{a_A}{C_2 b_A} \tag{18}$$

$$\frac{dx_3}{dt} = I_3 b_3 C_2 x_2 A - A_3 a_3 x_3 \quad x_{3,st} = \frac{C_1 I_1 b_1 b_P}{C_3 A_1 a_1 a_P} u \tag{19}$$

$$\frac{dP}{dt} = P(b_P C_1 x_1 - a_P) \quad P_{st} = \frac{A_2 a_2 b_P a_A}{C_2 I_2 b_2 a_P b_A} - \frac{T_P b_P}{b_2 a_P} \tag{20}$$

$$\frac{dA}{dt} = A(b_A C_2 x_2 - a_A) \quad A_{st} = \frac{C_1 I_1 A_3 b_1 a_3 b_P b_A}{A_1 C_3 I_3 a_1 b_3 a_P a_A} u \tag{21}$$

As long as the tumor secretion is below a threshold, $T_P < \frac{A_2 a_2 a_A}{C_2 I_2 b_A}$, the steady states of all HPA components stay fixed except for the pituitary (corticotroph) functional mass, which compensates and decreases with tumor secretion. This regime corresponds to the subclinical phase of Cushing's disease. When the tumor secretion is large enough to cross the threshold, $T_P > \frac{A_2 a_2 a_A}{C_2 I_2 b_A}$, the pituitary corticotroph functional mass goes to zero. In this limit, $P \to 0$, ACTH steady state depends only on tumor secretion, $x_{2,st} = \frac{I_2 T_P}{A_2 a_2} > \frac{a_A}{C_2 b_A}$ (from Eq. (18)). Substituting this lower limit of $x_{2,st}$ in Eq. (21) for adrenal growth, we obtain a positive net growth rate which means uncontrollable growth of the adrenal.

Limiting adrenal growth by a carrying capacity, $K_A$, stabilizes the system. Here is the steady-state solution of the pituitary adenoma system with an adrenal carrying capacity in the limit where $P \to 0$. In this regime, cortisol levels are high, and thus we assume $x_3 >> K_{GR}$:

$$\frac{dx_1}{dt} = I_1 b_1 \left(\frac{K_{GR}}{C_3 x_3}\right)^3 \frac{1}{C_3 x_3} u - A_1 a_1 x_1 \quad x_{1,st} = \frac{A_2^4 A_3^4 I_1 K_{GR}^3 a_2^4 a_3^4 b_1 b_A^4 u}{A_1 C_3^4 I_3^4 K_A^4 a_1 b_3^4 (C_2 I_2 T_P b_A - A_2 a_2 a_A)^4}$$
$$\tag{22}$$

$$\frac{dx_2}{dt} = I_2 T_P(t) - A_2 a_2 x_2 \quad x_{2,st} = \frac{I_2 T_P}{A_2 a_2} \tag{23}$$

$$\frac{dx_3}{dt} = I_3 b_3 C_2 x_2 A - A_3 a_3 x_3 \quad x_{3,st} = \frac{I_3 K_A b_3 (C_2 I_2 T_P b_A - A_2 a_2 a_A)}{A_2 A_3 a_2 a_3 b_A} \tag{24}$$

$$\frac{dA}{dt} = A\left(b_A C_2 x_2 \left(1 - \frac{A}{K_A}\right) - a_A\right) \quad A_{st} = K_A - \frac{A_2 K_A a_2 a_A}{C_2 I_2 T_P b_A} \tag{25}$$

We learn from the $x_{3,st}$ equation that interventions targeting ACTH or cortisol ($I_i, A_i, C_i; i = 1, 2$) alter cortisol steady state whereas CRH-targeting ($i = 1$) interventions do not in Cushing's disease.

Note that as long as the pituitary is not completely suppressed ($P > 0$), ACTH-targeting interventions would increase the subclinical–clinical threshold ($\frac{A_2 a_2 a_A}{C_2 I_2 b_A}$). This will cause the system dynamics to flow back towards the subclinical regime. This transition also happens with cortisol-targeting drugs when adding adrenal carrying capacity in the first model (before assuming $P \to 0$, Eqs. (17)–(21)). However, the steady state of that model is not analytically solvable. We show this behavior with numerical simulations (Fig. EV5).

**Adrenal adenoma:** We treat adrenal adenoma similarly to pituitary adenoma (Raz et al, 2023). We add the tumor cortisol secretion, $T_A(t)$, to Eq. (4):

$$\frac{dx_3}{dt} = I_3 b_3 C_2 x_2 A + I_3 T_A(t) - A_3 a_3 x_3 \tag{26}$$

In this case, the system has two steady states:

$$x_{1,st}^1 = \frac{a_P}{C_1 b_P} \quad x_{1,st}^2 = \frac{I_1 A_3 b_1 a_3 u}{A_1 C_3 I_3 a_1 T_A}$$

$$x_{2,st}^1 = \frac{a_A}{C_2 b_A} \quad x_{2,st}^2 = 0$$

$$x_{3,st}^1 = \frac{C_1 I_1 b_1 b_P}{A_1 C_3 a_1 a_P} u \quad x_{3,st}^2 = \frac{I_3 T_A}{a_3 A_3}$$

$$P_{st}^1 = \frac{A_2 a_2 b_P a_A}{C_2 I_2 b_2 a_P b_A} \quad P_{st}^2 = 0$$

$$A_{st}^1 = \frac{C_1 I_1 A_3 b_1 a_3 b_P b_A}{A_1 C_3 I_3 a_1 b_3 a_P a_A} u - \frac{T_A b_A}{a_A b_3} \quad A_{st}^2 = 0$$

As long as the adrenal tumor secretion is below a threshold, $T_A < \frac{C_1 I_1 A_3 b_1 a_3 b_P}{A_1 C_3 I_3 a_1 a_P} u$, the first steady state is the only stable point of the system. This regime is the subclinical phase where all the HPA variables remain normal due to the healthy adrenal cortex that shrinks and compensates. When the tumor secretion crosses that threshold, the adrenal cortex cannot compensate further and it goes to mass zero of the cortisol-releasing cells, $A \to 0$. Cortisol rises and inhibits CRH levels so that the net pituitary growth rate is negative and the pituitary mass goes to zero as well, $P \to 0$. The system is drawn toward the second steady state, which becomes stable. Cortisol steady state is determined by tumor secretion capacity. Thus, in this case of adrenal adenomas, only cortisol-targeting drugs are predicted to work in reducing the effects of hypercortisolism.

Cortisol-targeting drugs increase the subclinical–clinical threshold. If the glands are still able to recover, the system dynamics flow back to the subclinical regime. Note that CRH-modulating interventions could alter the threshold as well, however their effect will lead to an unwanted increase in long-term cortisol levels.

**HPA-targeting drug efficacy in nonadjustable gland model:** Here we consider a model with glands that cannot change their

**Table 3. Parameters of the HPA gland mass model.**

| Parameter | Value | Reference |
|---|---|---|
| $a_1$ | 0.17 [1/$min$] | Andersen et al, 2013 |
| $a_2$ | 0.035 [1/$min$] | Andersen et al, 2013 |
| $a_3$ | 0.0086 [1/$min$] | Andersen et al, 2013 |
| $a_P$ | 0.1 [1/$day$] | Karin et al, 2020 |
| $a_A$ | 0.05 [1/$day$] | Karin et al, 2020 |
| $K_{GR}$ | 4 | Karin et al, 2020 |
| $n$ | 3 | Karin et al, 2020 |

functional mass. Thus, we set $P = A = 1$ and the system becomes:

$$\frac{dx_1}{dt} = I_1 b_1 \frac{1}{C_3 x_3} \left(\frac{K_{GR}}{C_3 x_3}\right)^3 u - A_1 a_1 x_1 \quad x_{1,st} = \frac{A_2 A_3 C_3^3 a_2 a_3}{C_1 C_2 I_2 I_3 K_{GR}^3 b_2 b_3} \sqrt{\frac{C_1 C_2 I_1 I_2 I_3 K_{GR}^6 b_1 b_2 b_3 u}{A_1 A_2 A_3 C_3^7 a_1 a_2 a_3}}$$

$$\tag{27}$$

$$\frac{dx_2}{dt} = I_2 b_2 C_1 x_1 \left(\frac{K_{GR}}{C_3 x_3}\right)^3 - A_2 a_2 x_2 \quad x_{2,st} = \frac{A_3 a_3}{C_2 I_3 b_3} \sqrt[8]{\frac{C_1 C_2 I_1 I_2 I_3 K_{GR}^6 b_1 b_2 b_3 u}{A_1 A_2 A_3 C_3^7 a_1 a_2 a_3}}$$

$$\tag{28}$$

$$\frac{dx_3}{dt} = I_3 b_3 C_2 x_2 - A_3 a_3 x_3 \quad x_{3,st} = \sqrt[8]{\frac{C_1 C_2 I_1 I_2 I_3 K_{GR}^6 b_1 b_2 b_3 u}{A_1 A_2 A_3 C_3^7 a_1 a_2 a_3}}$$

$$\tag{29}$$

According to this model all points of interventions (all parameter changes) should work to alter cortisol steady state. A complementary numerical simulation is depicted in Fig. EV1.

### *Numerical simulations*

To analyze the model dynamics we used numerical simulations. Each simulation was initialized with the nominal HPA parameters (Karin et al, 2020) (Table 3) and a long warmup period of simulated time to reach the system's steady state. This state, at the end of the warmup, was defined as the system healthy baseline before perturbation simulations. Then we ran a specific perturbation for each pathological condition: increased stressor input for the chronic stress conditions and growing tumor for Cushing's syndrome. We simulated drugs as step functions of parameter change administered during the clinical stage with a dose that is predicted to cancel out the pathology effect, when possible, or at different doses as needed. The results presented in this study are fold-changes of the HPA variables with respect to the healthy baseline.

Numerical simulations were implemented in Python 3.9.7 using solvers of ordinary differential equations (ODEs), implemented in the scipy package version 1.7.3. (Virtanen et al, 2020).

## Data availability

Python code needed to reconstruct the analysis and figures is provided in the GitHub repository: https://github.com/tomermilo/hpa-drugs.

The source data of this paper are collected in the following database record: biostudies:S-SCDT-10_1038-S44320-024-00083-0.

## Peer review information

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

## Acknowledgements

The authors thank Dr. Jan Torleif Pedersen for fruitful discussions. This work was supported by a Cancer Research UK grant (C19767/A27145) and by the European Research Council under the European Union's Horizon 2020 research and innovation program (grant agreement No. 856487). The figures were created in part with the help of https://BioRender.com.

## Author contributions

**Tomer Milo**: Conceptualization; Software; Formal analysis; Validation; Investigation; Visualization; Methodology; Writing—original draft; Writing—review and editing. **Shiraz Nir Halber**: Conceptualization; Resources; Data curation; Investigation. **Moriya Raz**: Conceptualization; Investigation. **Dor Danan**: Validation; Investigation; Writing—review and editing. **Avi Mayo**: Supervision; Investigation; Methodology. **Uri Alon**: Conceptualization; Supervision; Funding acquisition; Investigation; Writing—original draft; Writing—review and editing.

Source data underlying figure panels in this paper may have individual authorship assigned. Where available, figure panel/source data authorship is listed in the following database record: biostudies:S-SCDT-10_1038-S44320-024-00083-0.

## Disclosure and competing interests statement

The authors declare no competing interests.

# Expanded View Figures

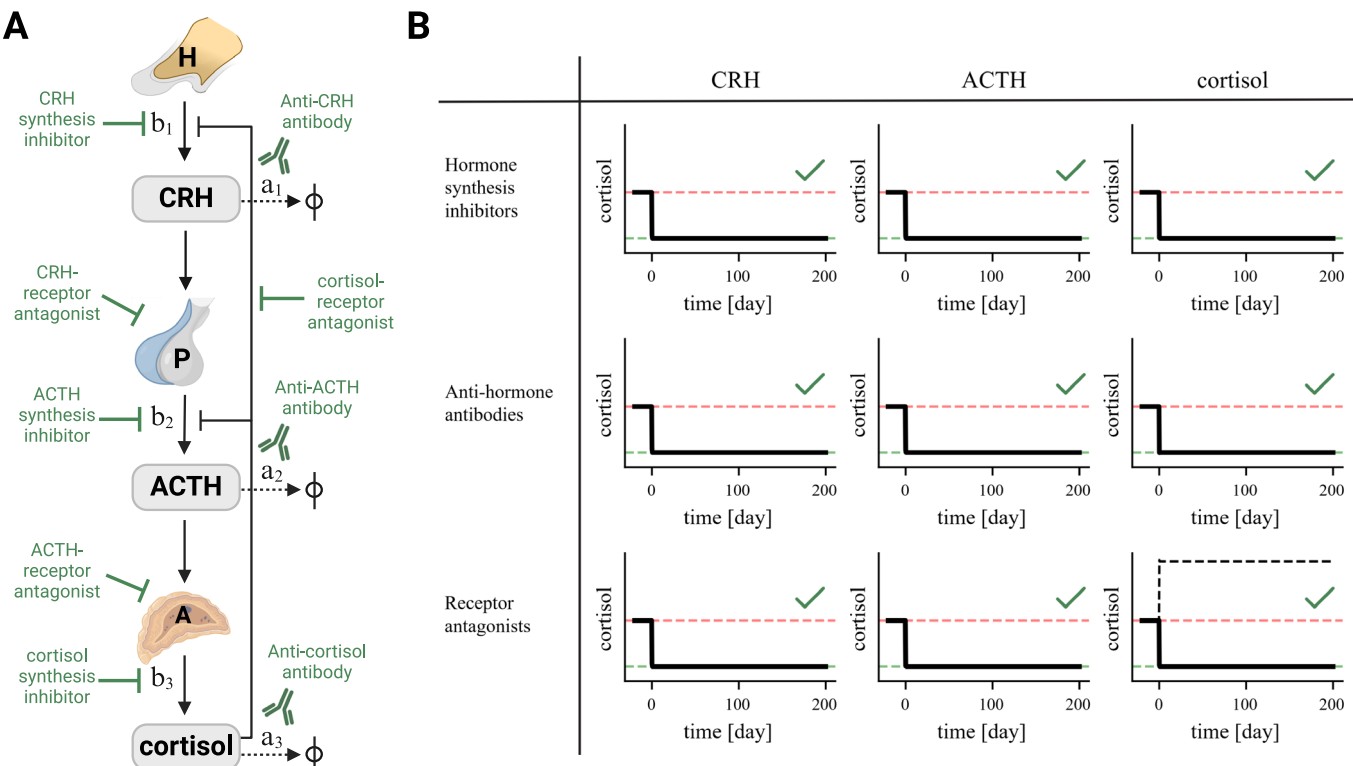

**Figure EV1.   Efficacy of potential interventions according to a HPA model with nonadjustable glands, related to Fig. 1.**

Panels similar to Fig. 1. (**A**) The classical HPA model with nonadjustable glands. (**B**) Simulations of HPA-modulating drugs' effect on cortisol levels under the classical HPA model.

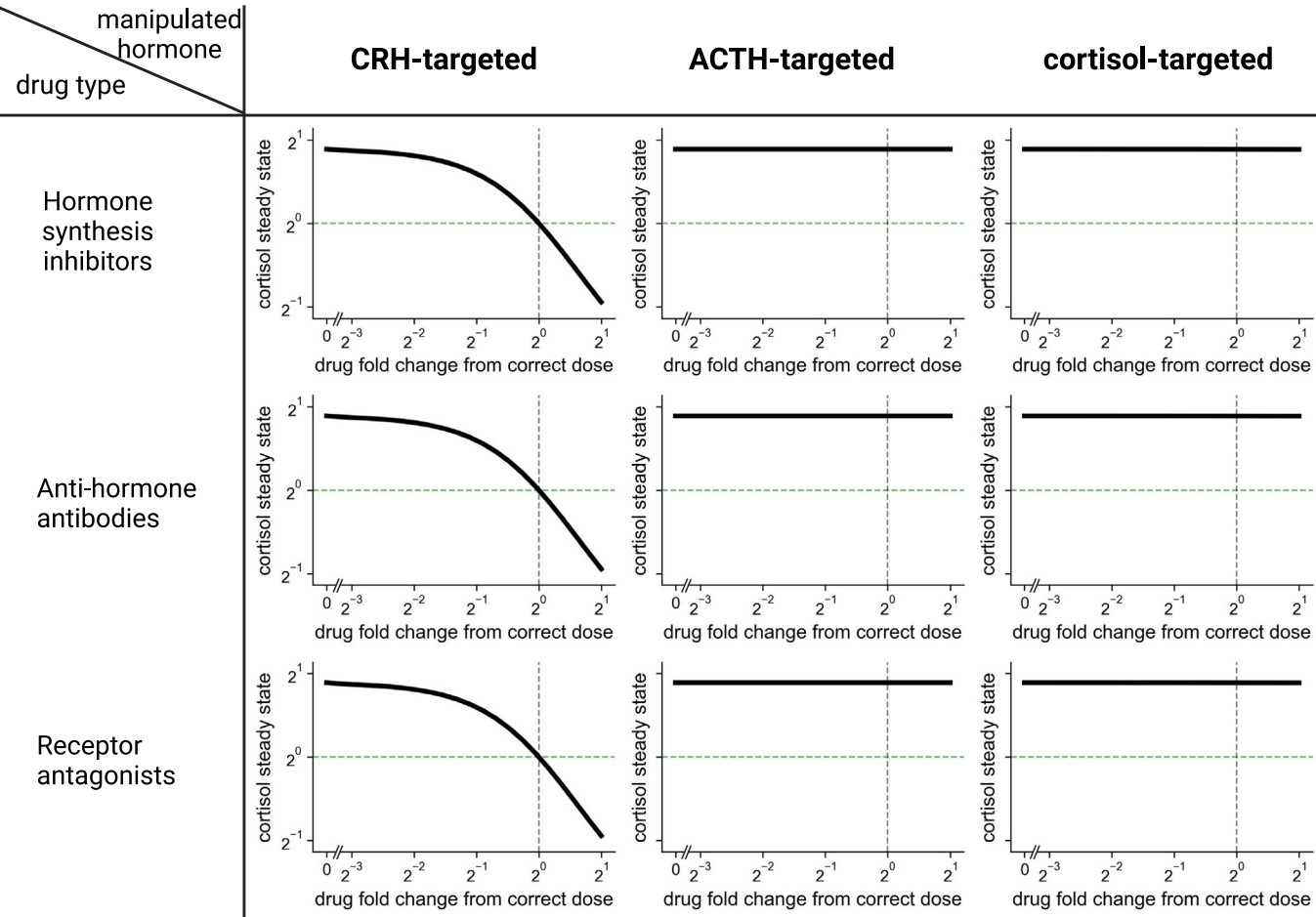

**Figure EV2.  Steady-state cortisol dose response to HPA-targeting drugs, related to Fig. 4A.**

Cortisol steady state in the model upon changes in doses of HPA-targeting drugs. Horizontal dashed green lines indicate normal cortisol steady-state level; Vertical dashed gray lines indicate the drug dose that normalizes cortisol.

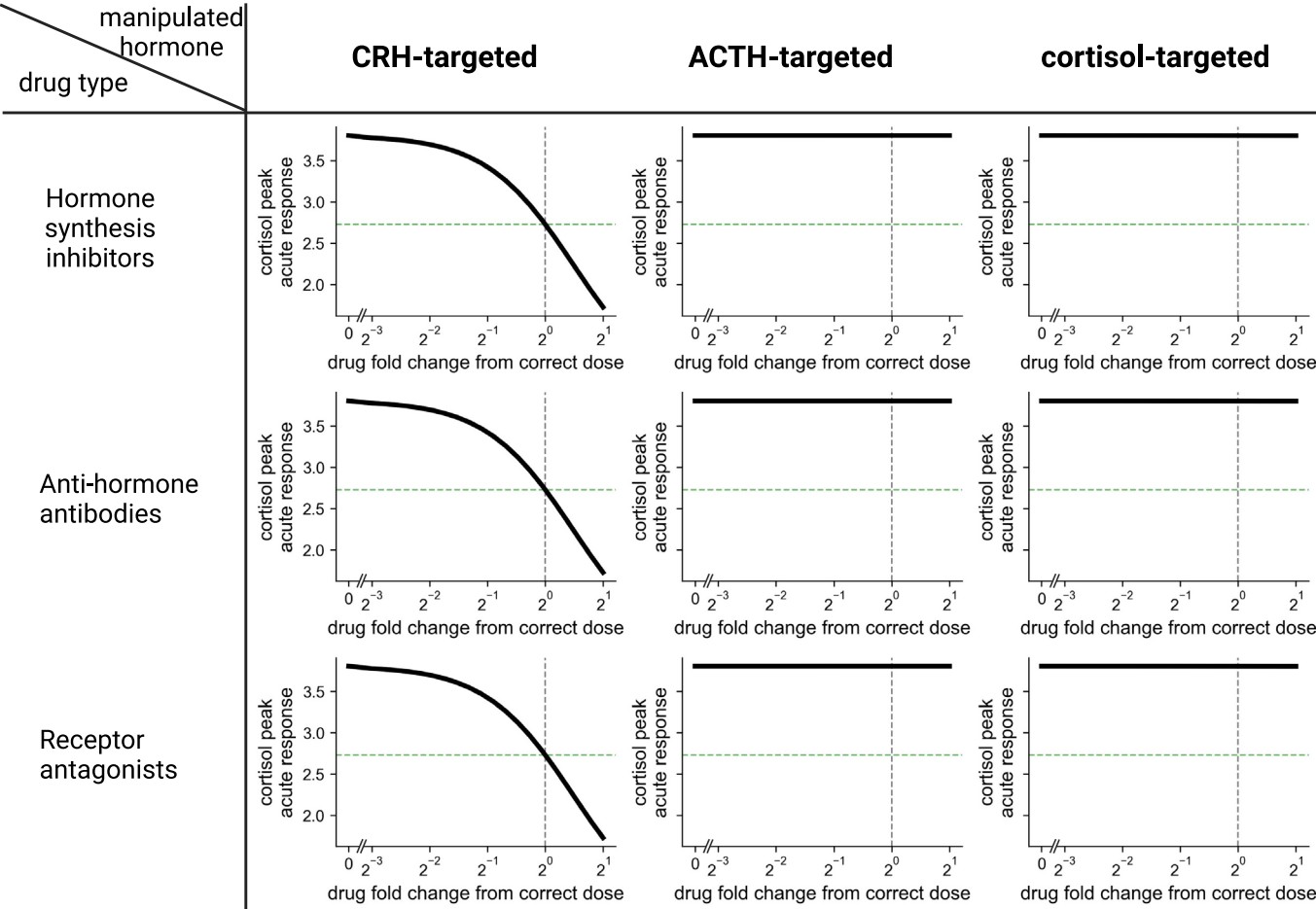

**Figure EV3. Cortisol peak response to acute stressor under varying concentrations of HPA-targeting drugs, related to Fig. 4D.**

Cortisol peak response to acute stress relative to steady state for different doses of HPA-targeting drugs. Horizontal dashed green lines indicate normal response; Vertical dashed gray lines indicate the drug dose that normalizes the response.

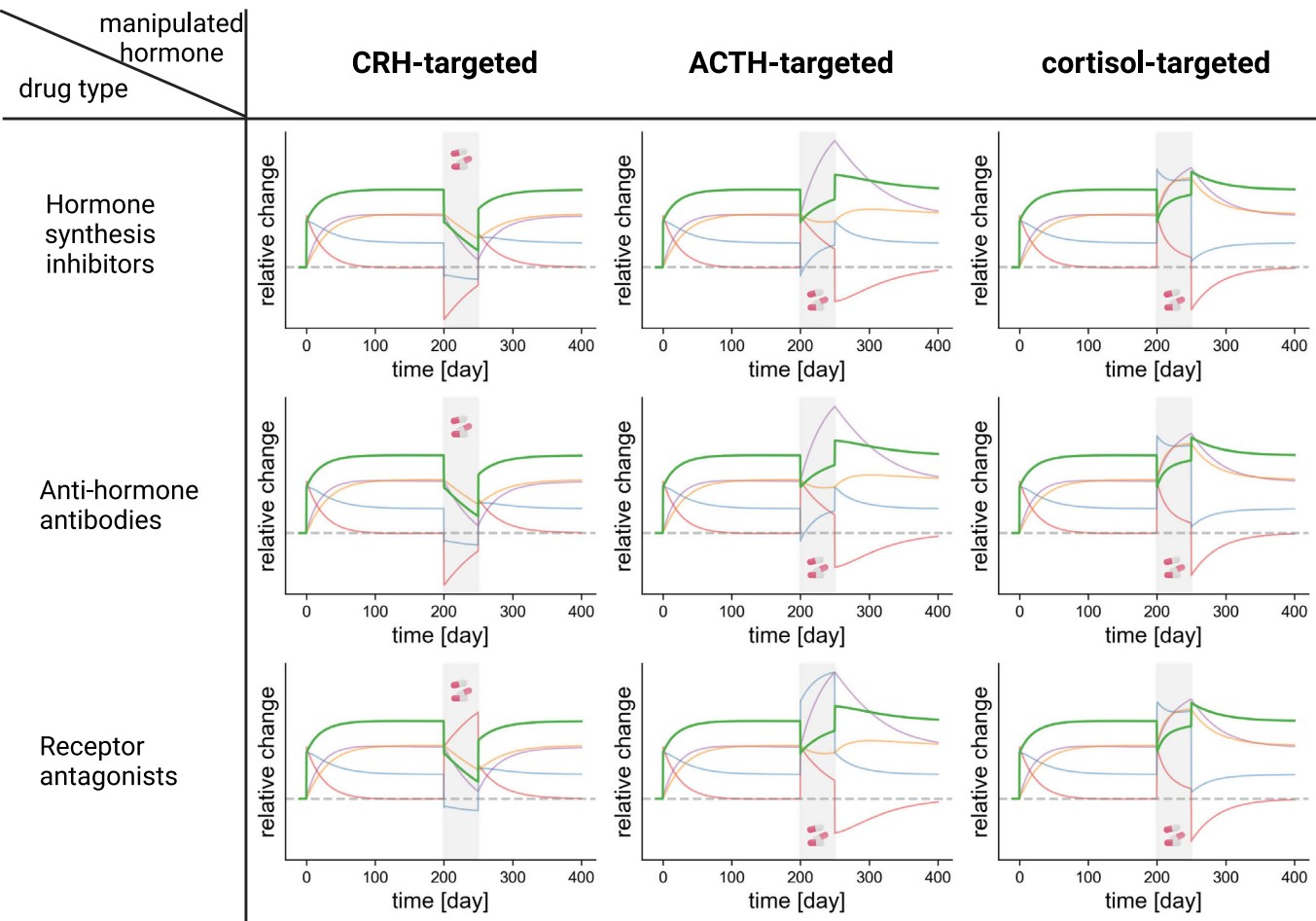

**Figure EV4.   Full HPA dynamics upon treatment cessation, related to Fig. 4E,F.**

HPA dynamics upon cessation of HPA-targeting drugs after 50 days. For color legend see Fig. 4E,F.

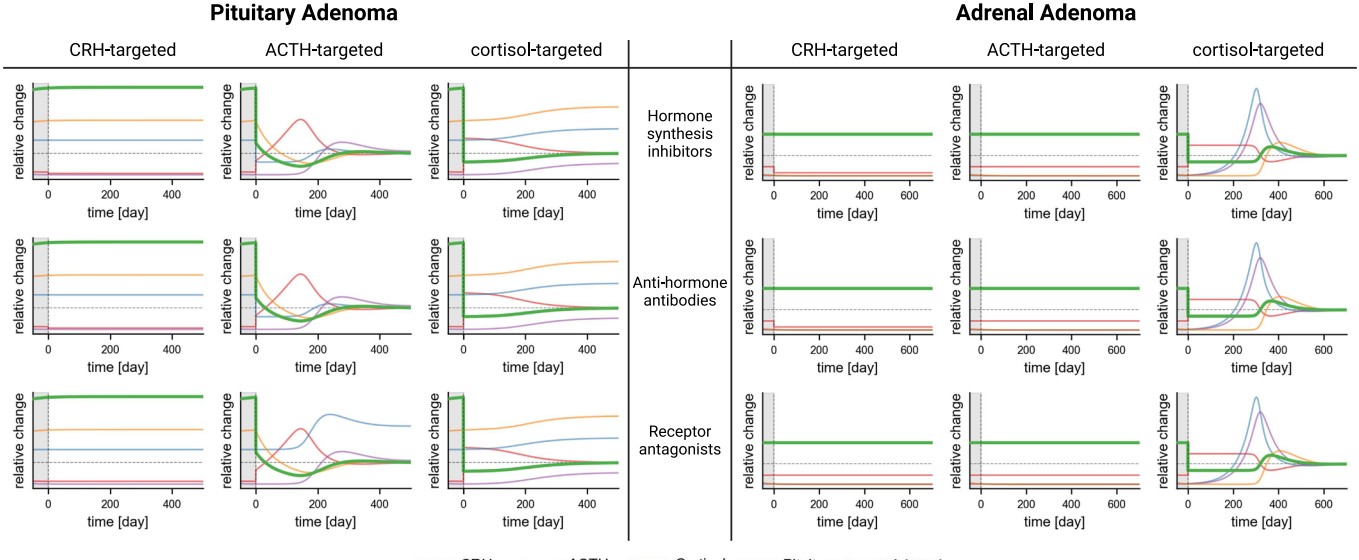

**Figure EV5.   HPA dynamics under HPA-targeting drugs in Cushing syndrome, related to Fig. 5.**

Simulations of the HPA dynamics during treating Cushing syndrome caused by a pituitary adenoma (left) or by an adrenal adenoma (right). The simulation starts with untreated Cushing (gray shaded region) and at point zero the simulated drug is administered.

