## [Peer Review File · Molecular Systems Biology]

Hormone circuit explains why most HPA drugs fail for mood disorders and predicts the few that work

Tomer Milo, Shiraz Nir Halber, Moriya Raz, Dor Danan, Avi Mayo, and Uri Alon

Corresponding author(s): Uri Alon (uri.alon@weizmann.ac.il)

Review Timeline:

Transfer from Review Commons:	24th Oct 24
Editorial Decision:	19th Nov 24
Revision Received:	2nd Dec 24
Accepted:	4th Dec 24

Editor: Jingyi Hou

Transaction Report: This manuscript was transferred to Molecular Systems Biology following peer review at Review Commons.

Review #1

1. Evidence, reproducibility and clarity:

Evidence, reproducibility and clarity (Required)

The authors address the question of lowering long-term elevated cortisol levels by affecting the parameters in a previously published mathematical model of the hypothalamic-pituitary-adrenal (HPA) axis. The parameters are related to various pathways. The elevation in cortisol levels is related to diseases e.g. mood disorders and Cushing's syndrome.

The authors conducted a systematic in silico analysis of various points of intervention in the HPA axis. They found that only two interventions targeting corticotropin-releasing hormone (CRH) can lower long-term cortisol. Other drug targets either fail to lower cortisol due to gland-mass compensation or lower cortisol but harm other aspects of the HPA axis. Thus, they identify potential drug targets, including CRH-neutralizing antibodies and CRH synthesis inhibitors, for lowering long-term cortisol in mood disorders and in those suffering from chronic stress.

The method used is in silico investigations of the mathematical model.

The draft is well written with a single typo in line 270. I have no further comments!

2. Significance:

Significance (Required)

In silico predictions without direct use of data is a weakness but the conducted analysis is convincing. An improved understanding of why some drugs work and others do not is important and is postulated to agree with clinical evidence.

3. How much time do you estimate the authors will need to complete the suggested revisions:

Estimated time to Complete Revisions (Required)

(Decision Recommendation)

Less than 1 month

4. Review Commons values the work of reviewers and encourages them to get credit for their work. Select 'Yes' below to register your reviewing activity at Web of Science Reviewer Recognition Service (formerly Publons); note that the content of your review will not be visible on Web of Science.

Yes

Review #2

1. Evidence, reproducibility and clarity:

Evidence, reproducibility and clarity (Required)

Summary

The authors utilise a mathematical model of the hypothalamic-pituitary-adrenal axis to address the utility of interventions altering its various outputs (CRH, ACTH and cortisol) to ameliorate axis disruption in response to chronic stress. They show that a lowering of circulating CRH by either blocking its synthesis or increasing its clearance is effective at returning the HPA axis to basal activity at all levels. In contrast, interventions altering ACTH or cortisol production, their circulating levels or actions are ineffective in the model. This is consistent with data on the long-term efficacy of drugs reducing excess corticosteroids in patients and animal models. The use of mathematical models to describe complex interactions in endocrine systems is a valuable advance in our understanding of potential mechanisms and therapies and this is an excellent example.

Major comments

1. The model of the HPA axis that the authors have described previously is a little simplistic when considering the known physiology. Specifically, this model ignores the contribution of vasopressin to the axis, which has been described as being the primary hypothalamic factor driving HPA axis activity in chronic stress (see [doi.org/10.1016/S0079-6123\(08\)00403-2](https://doi.org/10.1016/S0079-6123(08)00403-2)). Including this may be beyond the scope of the current model, however it should be considered and at least commented on. It is notable that the model fits the clinical and animal model data, which may suggest that the contribution of vasopressin in the long term may be overestimated, possibly as a result of differential effects of the two hypothalamic factors, with CRH driving ACTH release and POMC gene expression, whilst vasopressin only increases ACTH release without augmenting POMC expression. This is worthy of discussion.
2. The model that this study relies on is dependent on slow changes in the various levels of the endocrine axis and the authors have focused on alterations in cell number as the process leading to a prolongation of their dysfunction. For the stress axis, the evidence for changes in corticotroph cell number is weak and the recent paper of Lopez et al (DOI: 10.1126/sciadv.abe44) suggests that chronic stress, at least over a period of 3 weeks does not lead to an alteration in the number of corticotrophs, despite cell population changes in the adrenal gland. There are other processes which could lead to prolonged alteration of corticotroph output and it would be better to focus (as the authors have in places) on functional mass, rather than cell number which may suggest it is not the trophic effect of CRH that is important for increased functional mass.
3. The parameters in the model for interventions are described as simply being less than or greater than one- to what extent are the effects of these interventions dependent on their specific value? For example, presumably if the I1 value is close to zero, then the CRH-synthesis inhibitor would be ineffective. Likewise, if it were close to 1 then there would be negligible release of CRH in response to stress, and the preservation of a response to acute stress would be lost. Can the authors show the range of values for I1, C1 and A1 where the interventions are effective at normalising HPA axis function whilst (for I1 and A1) still preserving the acute stress response?

4. In the models that the authors describe with CRH interventions, what is the impact of stopping the intervention on axis output in the short and long-term? Presumably ceasing the use of CRH antagonists would lead to much more severe axis dysregulation than CRH neutralising antibodies or CRH synthesis inhibitors.

2. Significance:

Significance (Required)

Whilst the study builds on the use of a previously described mathematical model, its utility in identifying potential targets for therapy within the important area of chronic stress makes it an important example of the value of the modelling approach to decisions on appropriate targets for therapy. The model does not include important known factors which have been described as being important in the HPA axis response to chronic stress and would be considerably improved if these could be incorporated.

The study builds on conceptual insights into the role a delayed or slow functional mass change might play in dysregulation of endocrine axes and this could be applied to other physiological systems and will be of interest to modellers and physiologists alike. The authors are leaders in this field and there are few other modellers considering systems level interactions over this timescale.

As a pituitary physiologist, my review has focused on the interactions between the various players in HPA axis function, I do not have the expertise to comment on mathematical modelling aspects.

3. How much time do you estimate the authors will need to complete the suggested revisions:

Estimated time to Complete Revisions (Required)

(Decision Recommendation)

Less than 1 month

4. Review Commons values the work of reviewers and encourages them to get credit for their work. Select 'Yes' below to register your reviewing activity at Web of Science Reviewer Recognition Service (formerly Publons); note that the content of your review will not be visible on Web of Science.

Yes

Review #3

1. Evidence, reproducibility and clarity:

Evidence, reproducibility and clarity (Required)

This extremely interesting paper asks why various attempts to treat depression and bipolar disorder with glucocorticoid antagonists or cortisol synthesis inhibitors have failed. The starting point for their analysis is a simple computational model that, importantly, includes the facts that CRH stimulates not only ACTH release but also corticotroph growth and ACTH stimulates not only cortisol production but also the growth of cells in the adrenal cortex. They call this the "gland mass model". According to the model, if the hypothalamus receives a continuous stress input, all of the HPA hormones will be elevated-CRH transiently and the others in a sustained fashion. Adding a sufficient dose of a CRH inhibitor (decreasing the rate constant b_1 in the model) or a CRH antibody (increasing the rate constant a_1) normalizes the hormone levels, whereas blocking cortisol function or production does not. This is demonstrated by numerical simulations and backed up by deriving analytical expressions for the hormone concentrations at steady state. The paper provides a plausible explanation for why past therapeutic efforts have failed and points to a couple of approaches that might succeed. These conclusions are hypotheses-they haven't been tested experimentally and we really don't know how accurately the system is described by this nice, simple model-but they are really intriguing hypotheses that could lead to therapeutic breakthroughs. I strongly recommend publication.

****My only criticisms are minor:****

1. The authors should specify what exact change in the model's parameters they are making to implement their therapeutic interventions. E.g. in Fig 1B top left and 2A, what is the change in the value of b_1 that corresponds to the addition of a CRH-synthesis inhibitor? (I'd guess it's being dropped to zero, but if this is stated, I missed it)
2. I think it would also be useful to show a dose-response relationship for the various interventions.

****Referees cross-commenting****

It looks like we are all enthusiastic about this work.

2. Significance:

Significance (Required)

Strengths: It's a beautiful new insight on a really important topic, extracted from a simple, understandable mathematical model of the HPA axis.

Weaknesses: It is based on a model and the model could be wrong. This does not however diminish my enthusiasm for this provocative work.

Advance: It is highly original.

Audience: I hope attracts a wide audience--modelers, endocrinologists, psychiatrists, drug developers.

My expertise: I am a systems biologist, have taught psychopharmacology to medical students, and have an interest in endocrine signaling.

3. How much time do you estimate the authors will need to complete the suggested revisions:

Estimated time to Complete Revisions (Required)

(Decision Recommendation)

Less than 1 month

Yes

Full Revision

Manuscript number: RC-2024-02578

Corresponding author(s): Uri Alon

1. General Statements

The reviewer comments helped us improve the paper by including new computations, figures, and analyses related to vasopressin, drug dosages, and treatment cessation. We have also removed confusing terminology from the text. We believe that the paper is now more comprehensive, clear, and rigorous.

Reviewer #1 (Evidence, reproducibility and clarity (Required)):

The authors address the question of lowering long-term elevated cortisol levels by affecting the parameters in a previously published mathematical model of the hypothalamic-pituitary-adrenal (HPA) axis. The parameters are related to various pathways. The elevation in cortisol levels is related to diseases e.g. mood disorders and Cushing's syndrome.

The authors conducted a systematic in silico analysis of various points of intervention in the HPA axis. They found that only two interventions targeting corticotropin-releasing hormone (CRH) can lower long-term cortisol. Other drug targets either fail to lower cortisol due to gland-mass compensation or lower cortisol but harm other aspects of the HPA axis. Thus, they identify potential drug targets, including CRH-neutralizing antibodies and CRH synthesis inhibitors, for lowering long-term cortisol in mood disorders and in those suffering from chronic stress.

The method used is in silico investigations of the mathematical model.

The draft is well written with a single typo in line 270. I have no further comments!

Response: The typo is fixed.

Reviewer #1 (Significance (Required)):

In silico predictions without direct use of data is a weakness but the conducted analysis is convincing. An improved understanding of why some drugs work and others do not is important and is postulated to agree with clinical evidence.

Response: We thank the reviewer for this endorsement.

Reviewer #2 (Evidence, reproducibility and clarity (Required)):

Summary

The authors utilise a mathematical model of the hypothalamic-pituitary-adrenal axis to address the utility of interventions altering its various outputs (CRH, ACTH and cortisol) to ameliorate axis disruption in response to chronic stress. They show that a lowering of circulating CRH by either blocking its synthesis or increasing its clearance is effective at returning the HPA axis to basal activity at all levels. In contrast, interventions altering ACTH or cortisol production, their circulating levels or actions are ineffective in the model. This is consistent with data on the long-term efficacy of drugs reducing excess corticosteroids in patients and animal models. The use of mathematical models to describe complex interactions in endocrine systems is a valuable advance in our understanding of potential mechanisms and therapies and this is an excellent example.

Response: We thank the reviewer for this endorsement.

Major comments

1. The model of the HPA axis that the authors have described previously is a little simplistic when considering the known physiology. Specifically, this model ignores the contribution of vasopressin to the axis, which has been described as being the primary hypothalamic factor driving HPA axis activity in chronic stress (see doi.org/10.1016/S0079-6123(08)00403-2). Including this may be beyond the scope of the current model, however it should be considered and at least commented on. It is notable that the model fits the clinical and animal model data, which may suggest that the contribution of vasopressin in the long term may be overestimated, possibly as a result of differential effects of the two hypothalamic factors, with CRH driving ACTH release and POMC gene expression, whilst vasopressin only increases ACTH release without augmenting POMC expression. This is worthy of discussion.

Response: We thank the reviewer for this comment which helped us discuss vasopressin. We agree that adding it as a variable in the model is beyond the scope of the current study. We describe its effects in the introduction and discussion sections. Interestingly, when one considers the best characterized effect of vasopressin, namely enhancing CRH-dependent ACTH release, one can use this model to investigate the effects of inhibiting vasopressin. We predict that vasopressin inhibition is unlikely to be an effective strategy for lowering long-term cortisol and alleviating stress-related mental disorders, as evidenced by the failure of clinical trials.

In the introduction we add:

1. "CRH stimulates the secretion of adrenocorticotrophic hormone (ACTH) by corticotroph cells in the anterior pituitary, an effect enhanced by vasopressin (Aguilera et al, 2008; Antoni, 2017)." (lines 34-36)
2. Clinical trials for two vasopressin 1b receptor antagonist candidates, SSR149415 and TS-121, in the table of HPA-related clinical trials (Table 1)

In the discussion we add (lines 398-409): "One important factor not explicitly considered in the model is the contribution of vasopressin to the axis. Vasopressin potentiates the CRH-dependent release of ACTH from pituitary corticotrophs by acting on the V1b receptor (V1bR) (Aguilera et al, 2008; Antoni, 2017). Including this hormone explicitly is beyond the current scope. However, a simple analysis indicates that the effect of elevated vasopressin can be modeled by increasing the ACTH secretion parameter b_2 . This suggests that vasopressin V1b receptor antagonists should have effects similar to inhibitors of ACTH production. As such, vasopressin receptor antagonists should be compensated by the HPA axis without long-term effects on cortisol. Accordingly, V1bR antagonists did not show statistically significant efficacy in clinical trials for major depressive disorder and generalized anxiety disorder (Griebel et al, 2012; Chaki, 2021; Kamiya et al, 2020). However, vasopressin may have additional relevant effects on the HPA axis and the central nervous system which warrant a more detailed modeling analysis."

2. The model that this study relies on is dependent on slow changes in the various levels of the endocrine axis and the authors have focused on alterations in cell number as the process leading to a prolongation of their dysfunction. For the stress axis, the evidence for changes in corticotroph cell number is weak and the recent paper of Lopez et al (DOI: 10.1126/sciadv.abe44) suggests that chronic stress, at least over a period of 3 weeks does not lead to an alteration in the number of corticotrophs, despite cell population changes in the adrenal gland. There are other processes which could lead to prolonged alteration of corticotroph output and it would be better to focus (as the authors have in places) on functional mass, rather than cell number which may suggest it is not the trophic effect of CRH that is important for increased functional mass.

Response: We thank the reviewer for this. We now refer only to functional mass changes. We corrected all places in which hyperplasia of corticotrophs is mentioned. We also state in lines 124-125 that the model is agnostic as to whether growth in functional mass is due to hyperplasia or hypertrophy.

We also added a citation for Lopez et al. 2021 (line 85) to support the growth of cortisol-secreting cells in the zona fasciculata of the adrenal gland under stress conditions.

3. The parameters in the model for interventions are described as simply being less than or greater than one- to what extent are the effects of these interventions dependent on their specific value? For example, presumably if the I_1 value is close to zero, then the CRH-synthesis inhibitor would be ineffective. Likewise, if it were close to 1 then there would be negligible release of CRH in response to stress, and the preservation of a response to acute stress would be lost. Can the authors show the range of values for I_1 , C_1 and A_1 where the interventions are effective at normalising HPA axis function whilst (for I_1 and A_1) still preserving the acute stress response?

Response: We thank the reviewer for this comment that helped us to add a new section in the results on dose response, and three new figures (Figure 4, Figure S2 and Figure S3):

“CRH interventions have a dose-dependent response in the model

We computed the effects of drug doses by varying the relevant model parameter, where zero dose means no change in the parameter and high doses mean large changes in the parameter. We find that both candidate interventions for lowering cortisol - CRH-synthesis inhibitors and CRH-blocking antibodies - cause a dose-dependent reduction of steady-state cortisol (Figure 4A). This indicates that putative treatment may require finding the appropriate dose to return the patients to their normal cortisol baseline range. Other drug candidates have no effect on long-term cortisol steady state (Figure S2).

At all doses, the steady states of CRH and ACTH remain normal (Figure 4B-C). The acute stress response, defined as peak cortisol upon acute stress input relative to steady-state cortisol, is dose dependent (Figure 4D and Figure S3). At a dose that returns cortisol to the normal range, the acute response is also normalized.

We also tested the effects of abrupt treatment cessation. For both CRH interventions, stopping treatment led to a rapid return to hypercortisolemia (Figure 4E-F and Figure S4).

Figure 4. Predicted effective interventions have a dose-dependent effect on cortisol, and cortisol abruptly rises when treatment is ceased. (A) Cortisol steady state in the model upon changes in doses of CRH-synthesis inhibitors and CRH-blocking antibodies. **(B-C)** The same changes in drug doses have no effect on ACTH **(B)** and CRH **(C)** steady state levels. **(D)** Cortisol peak response to an acute stress relative to steady state for different drug doses. **(E-F)** HPA dynamics upon cessation of CRH-synthesis inhibitors **(E)** and anti-CRH antibodies **(F)** after 50 days.”

In the supplemental information:

“Cortisol dose response to HPA-targeting drugs

Figure S2. Cortisol steady state dose response to HPA-targeting drugs, related to Figure 4.

Figure S3. Cortisol peak response to acute stressor under varying concentrations of HPA-targeting drugs, related to Figure 4.”

4. In the models that the authors describe with CRH interventions, what is the impact of stopping the intervention on axis output in the short and long-term? Presumably ceasing the use of CRH antagonists would lead to much more severe axis dysregulation than CRH neutralising antibodies or CRH synthesis inhibitors.

Response: We have now added new analysis on drug cessation (new figure 4E-F, Figure S4). After a 50 day treatment, sudden cessation caused a rapid return to hypercortisolemia: We added in lines 276-277: “We also tested the effects of abrupt treatment cessation. For both CRH interventions, stopping treatment led to a rapid return to hypercortisolemia (Figure 4E-F and Figure S4).”

(E-F) HPA dynamics upon cessation of CRH-synthesis inhibitors (E) and anti-CRH antibodies (F) after 50 days.

Figure S4:

Reviewer #2 (Significance (Required)):

Full Revision

Whilst the study builds on the use of a previously described mathematical model, its utility in identifying potential targets for therapy within the important area of chronic stress makes it an important example of the value of the modelling approach to decisions on appropriate targets for therapy. The model does not include important known factors which have been described as being important in the HPA axis response to chronic stress and would be considerably improved if these could be incorporated.

The study builds on conceptual insights into the role a delayed or slow functional mass change might play in dysregulation of endocrine axes and this could be applied to other physiological systems and will be of interest to modellers and physiologists alike. The authors are leaders in this field and there are few other modellers considering systems level interactions over this timescale.

Response: We thank the reviewer for this endorsement.

As a pituitary physiologist, my review has focused on the interactions between the various players in HPA axis function, I do not have the expertise to comment on mathematical modelling aspects.

Reviewer #3 (Evidence, reproducibility and clarity (Required)):

This extremely interesting paper asks why various attempts to treat depression and bipolar disorder with glucocorticoid antagonists or cortisol synthesis inhibitors have failed. The starting point for their analysis is a simple computational model that, importantly, includes the facts that CRH stimulates not only ACTH release but also corticotroph growth and ACTH stimulates not only cortisol production but also the growth of cells in the adrenal cortex. They call this the "gland mass model". According to the model, if the hypothalamus receives a continuous stress input, all of the HPA hormones will be elevated-CRH transiently and the others in a sustained fashion. Adding a sufficient dose of a CRH inhibitor (decreasing the rate constant b_1 in the model) or a CRH antibody (increasing the rate constant a_1) normalizes the hormone levels, whereas blocking cortisol function or production does not. This is demonstrated by numerical simulations and backed up by deriving analytical expressions for the hormone concentrations at steady state. The paper provides a plausible explanation for why past therapeutic efforts have failed and points to a couple of approaches that might succeed. These conclusions are hypotheses-they haven't been tested experimentally and we really don't know how accurately the system is described by this nice, simple model-but they are really intriguing hypotheses that could lead to therapeutic breakthroughs. I strongly recommend publication.

Response: We thank the reviewer for this endorsement.

My only criticisms are minor:

1. The authors should specify what exact change in the model's parameters they are making to implement their therapeutic interventions. E.g. in Fig 1B top left and 2A, what is the change in the value of b_1 that corresponds to the addition of a CRH-synthesis inhibitor? (I'd guess it's being dropped to zero, but if this is stated, I missed it)

Response: We thank the reviewer for that comment which helped us to clarify what is the required parameter change to normalize cortisol. We have now added in lines 172-174: "According to equation (1), as a general guideline, treating cortisol levels that are x-fold higher than baseline requires a drug dose that alters the relevant parameter (e.g., CRH production or removal rate) by a similar x-fold."

2. I think it would also be useful to show a dose-response relationship for the various interventions.

Response: We thank the reviewer for this comment that helped us to add a new section in the results on dose response, and three new figures (Figure 4, Figure S2 and Figure S3):

"CRH interventions have a dose-dependent response in the model"

We computed the effects of drug doses by varying the relevant model parameter, where zero dose means no change in the parameter and high doses mean large changes in the parameter. We find that both candidate interventions for lowering cortisol - CRH-synthesis inhibitors and

CRH-blocking antibodies - cause a dose-dependent reduction of steady-state cortisol (Figure 4A). This indicates that putative treatment may require finding the appropriate dose to return the patients to their normal cortisol baseline range. Other drug candidates have no effect on long-term cortisol steady state (Figure S2).

At all doses, the steady states of CRH and ACTH remain normal (Figure 4B-C). The acute stress response, defined as peak cortisol upon acute stress input relative to steady-state cortisol, is dose dependent (Figure 4D and Figure S3). At a dose that returns cortisol to the normal range, the acute response is also normalized.

We also tested the effects of abrupt treatment cessation. For both CRH interventions, stopping treatment led to a rapid return to hypercortisolemia (Figure 4E-F and Figure S4).

Figure 4. Predicted effective interventions have a dose-dependent effect on cortisol, and cortisol abruptly rises when treatment is ceased. (A) Cortisol steady state in the model upon changes in doses of CRH-synthesis inhibitors and CRH-blocking antibodies. **(B-C)** The same changes in drug doses have no effect on ACTH **(B)** and CRH **(C)** steady state levels. **(D)** Cortisol peak response to an acute stress relative to steady state for different drug doses. **(E-F)** HPA dynamics upon cessation of CRH-synthesis inhibitors **(E)** and anti-CRH antibodies **(F)** after 50 days.”

In the supplemental information:

“Cortisol dose response to HPA-targeting drugs

Figure S2. Cortisol steady state dose response to HPA-targeting drugs, related to Figure 4.

Figure S3. Cortisol peak response to acute stressor under varying concentrations of HPA-targeting drugs, related to Figure 4.”

Referees cross-commenting

It looks like we are all enthusiastic about this work.

Response: Thank you.

Reviewer #3 (Significance (Required)):

Strengths: It's a beautiful new insight on a really important topic, extracted from a simple, understandable mathematical model of the HPA axis.

Weaknesses: It is based on a model and the model could be wrong. This does not however diminish my enthusiasm for this provocative work.

Advance: It is highly original.

Full Revision

Audience: I hope attracts a wide audience--modelers, endocrinologists, psychiatrists, drug developers.

My expertise: I am a systems biologist, have taught psychopharmacology to medical students, and have an interest in endocrine signaling.

19th Nov 2024

Manuscript Number: MSB-2024-12710-T

Title: Hormone circuit explains why most HPA drugs fail for mood disorders and predicts the few that work

Author: Tomer Milo

Shiraz Nir Halber

Moriya Raz

Dor Danan

Avi Mayo

Uri Alon

Dear Uri,

Thank you for the submission of your revised manuscript to Molecular Systems Biology. We have now received the enclosed reports from the two reviewers who agreed to re-assess it. As you will see, the reviewers are both supportive, and I am pleased to inform you that we will be able to accept your manuscript pending the following amendments:

1. The manuscript file should be uploaded as a .docx. Figures should be removed from the manuscript and uploaded as individual figure files.
2. Please provide up to five keywords.
3. Please remove the Authors' Contribution section from the manuscript file.
4. At EMBO Press we ask authors to provide source data for the main figures. Our source data coordinator will contact you to discuss which figure panels we would need source data for and will also provide you with helpful tips on how to upload and organize the files.
5. Please include funding information in both the manuscript file and the online submission system, when applicable.
6. The "Conflict of interest" statement should be renamed to "Disclosure and competing interests statement".
7. All Materials and Methods need to be described in the main text using our 'Structured Methods' format, which is required for all research articles. According to this format, the Methods section includes a Reagents and Tools Table (listing key reagents, experimental models, software and relevant equipment and including their sources and relevant identifiers) followed by a Methods and Protocols section describing the methods using a step-by-step protocol format. The aim is to facilitate adoption of the methodologies across labs.

Please download and fill our Reagents and Tools Table template (.docx), which you can find in our author guidelines: <https://www.embopress.org/page/journal/17444292/authorguide#structuredmethods>

8. DATA availability section needs to be renamed to "Data Availability" instead of "Code and data Availability".
9. Appendix:
 - The appendix file should be in PDF format;
 - the "Supplemental information" should be removed from manuscript file and uploaded as an individual Appendix PDF. The Appendix should include a title page containing 'Appendix for + manuscript title' and a 'Table of content' listing the items with corresponding page numbers;
 - nomenclature should be 'Appendix Figure Sx' throughout the manuscript and Appendix PDF.
10. Please provide a "standfirst text" summarizing the study in one or two sentences (approximately 250 characters, including space), three to four "bullet points" highlighting the main findings and a "synopsis image" (550px width and 400-600 px height, PNG format) to highlight the paper on our homepage.

Here are a couple of examples:

<https://www.embopress.org/doi/10.15252/msb.20199356>

<https://www.embopress.org/doi/10.15252/msb.20209475>

<https://www.embopress.org/doi/10.15252/msb.209495>

11. Section order should be corrected: Title page - Abstract & Keywords - Introduction - Results - Discussion - Methods - Data Availability - Acknowledgements - Disclosure and Competing Interests Statement - References - Figure Legends - Table(s) - Expanded View Figure Legends

12. When you resubmit your manuscript, please download our CHECKLIST (<https://www.embopress.org/pb-assets/embo-site/EMBO%20Press%20Author%20Checklist-1642513524327.xlsx>) and include the completed form in your submission. *Please note* that the Author Checklist will be published alongside the paper as part of the transparent process (<https://www.embopress.org/page/journal/17444292/authorguide#transparentprocess>).

Click on the link below to submit your revised paper.

Link Not Available

Kind regards,
Jingyi

Jingyi Hou, PhD
Scientific Editor
Molecular Systems Biology

*** PLEASE NOTE *** As part of the EMBO Publications transparent editorial process initiative (see our Editorial at <https://www.nature.com/msb/journal/v6/n1/full/msb201072.html>), Molecular Systems Biology will publish online a Review Process File to accompany accepted manuscripts. When preparing your letter of response, please be aware that in the event of acceptance, your cover letter/point-by-point document will be included as part of this File, which will be available to the scientific community. More information about this initiative is available in our Instructions to Authors. If you have any questions about this initiative, please contact the editorial office (msb@embo.org).

Reviewer #1:

The authors utilise a mathematical model of the hypothalamic-pituitary-adrenal axis to address the utility of interventions altering its various outputs (CRH, ACTH and cortisol) to ameliorate axis disruption in response to chronic stress. They show that a lowering of circulating CRH by either blocking its synthesis or increasing its clearance is effective at returning the HPA axis to basal activity at all levels. In contrast, interventions altering ACTH or cortisol production, their circulating levels or actions are ineffective in the model. This is consistent with data on the long-term efficacy of drugs reducing excess corticosteroids in patients and animal models.

The use of mathematical models to describe complex interactions in endocrine systems is a valuable advance in our understanding of potential mechanisms and therapies and this is an excellent example. This is exemplified by the ability of the model to show the dose-response effects of intervention, which could be built into therapeutic protocols for treatment optimisation.

Reviewer #2:

With these revisions, especially the new Fig 4, the authors have made their strong and very interesting paper better. I support publication.

Rev_Com_number: RC-2024-02578

New_manu_number: MSB-2024-12710-T

Corr_author: Alon

Title: Hormone circuit explains why most HPA drugs fail for mood disorders and predicts the few that work

Dear Uri,

Thank you for the submission of your revised manuscript to Molecular Systems Biology. We have now received the enclosed reports from the two reviewers who agreed to re-assess it. As you will see, the reviewers are both supportive, and I am pleased to inform you that we will be able to accept your manuscript pending the following amendments:

Response: Thank you

1. The manuscript file should be uploaded as a .docx. Figures should be removed from the manuscript and uploaded as individual figure files.

Response: Done

2. Please provide up to five keywords.

Response: Done

3. Please remove the Authors' Contribution section from the manuscript file.

Response: We validated that there is no Authors' Contribution section in the manuscript file

4. At EMBO Press we ask authors to provide source data for the main figures. Our source data coordinator will contact you to discuss which figure panels we would need source data for and will also provide you with helpful tips on how to upload and organize the files.

Response: The data coordinator approved our source data

5. Please include funding information in both the manuscript file and the online submission system, when applicable.

Response: Done

6. The "Conflict of interest" statement should be renamed to "Disclosure and competing interests statement".

Response: Done

7. All Materials and Methods need to be described in the main text using our 'Structured Methods' format, which is required for all research articles. According to this format, the Methods section includes a Reagents and Tools Table (listing key reagents, experimental models, software and relevant equipment and including their sources and relevant identifiers) followed by a Methods and Protocols section describing the methods using a step-by-step protocol format. The aim is to facilitate adoption of the methodologies across labs.

Please download and fill our Reagents and Tools Table template (.docx), which you can find in our author guidelines:

<https://www.embopress.org/page/journal/17444292/authorguide#structuredmethods>

Response: Done

8. DATA availability section needs to be renamed to "Data Availability" instead of "Code and data Availability".

Response: Done

9. Appendix:

- The appendix file should be in PDF format;
- the "Supplemental information" should be removed from manuscript file and uploaded as an individual Appendix PDF. The Appendix should include a title page containing 'Appendix for + manuscript title' and a 'Table of content' listing the items with corresponding page numbers;
- nomenclature should be 'Appendix Figure Sx' throughout the manuscript and Appendix PDF.

Response: We now embedded the Supplementary Information in the main text. We moved the mathematical part to the methods section and changed the figures to be Expanded View Figures. We no longer have an Appendix / Supplemental information section.

10. Please provide a "standfirst text" summarizing the study in one or two sentences (approximately 250 characters, including space), three to four "bullet points" highlighting the main findings and a "synopsis image" (550px width and 400-600 px height, PNG format) to highlight the paper on our homepage.

Here are a couple of examples:

<https://www.embopress.org/doi/10.15252/msb.20199356>

<https://www.embopress.org/doi/10.15252/msb.20209475>

<https://www.embopress.org/doi/10.15252/msb.209495>

Response: Done

11. Section order should be corrected: Title page - Abstract & Keywords - Introduction - Results - Discussion - Methods - Data Availability - Acknowledgements - Disclosure and Competing Interests Statement - References - Figure Legends - Table(s) - Expanded View Figure Legends

Response: Done

12. When you resubmit your manuscript, please download our CHECKLIST (<https://www.embopress.org/pb-assets/embo-site/EMBO%20Press%20Author%20Checklist-1642513524327.xlsx>) and include the completed form in your submission.

Please note that the Author Checklist will be published alongside the paper as part of the transparent process (<https://www.embopress.org/page/journal/17444292/authorguide#transparentprocess>).

Response: Done

Click on the link below to submit your revised paper.

<https://msb.msubmit.net/cgi-bin/main.plex?el=A5lk6BEIw2A2FaH2I7A9ftdQ73Ix3Ljqny5jyrcScqSgY>

Kind regards,
Jingyi

Jingyi Hou, PhD
Scientific Editor
Molecular Systems Biology

4th Dec 2024

Manuscript number: MSB-2024-12710R

Title: Hormone circuit explains why most HPA drugs fail for mood disorders and predicts the few that work

Dear Uri,

Thank you again for sending us your revised manuscript. We are now satisfied with the modifications made and I am pleased to inform you that your paper has been accepted for publication.

Yours sincerely,

Sincerely,
Jingyi

Jingyi Hou, PhD
Scientific Editor
Molecular Systems Biology
